# FROM OBJECTS TO SKILLS: INTERPRETABLE META-POLICIES FOR NEURAL CONTROL

## ABSTRACT

Despite its success in learning high-performing policies for diverse control and decision-making tasks, deep reinforcement learning remains difficult to interpret and align due to the black-box nature of its neural network representations. Neuro-symbolic approaches improve transparency by incorporating symbolic reasoning, but when applied to low-level actions, they result in overly complex policies. We introduce NEXUS, a hierarchical Reinforcement Learning framework that integrates neural skills with neuro-symbolic meta-policies to balance efficiency and interpretability. In its core, it allows transparent reasoning on disentangled high-level actions (i.e. interpretable skills), greatly reducing complexity of symbolic policies. Object-centric representations enable extracting rewards and meta-policies from language models, while the hierarchical structure allow reasoning over skills rather than atomic actions. We experimentally demonstrate that NEXUS agents are interpretable, less prone to reward hacking, and more robust to environment simplifications. We further evaluate how differing levels of meta-policy interpretability (i.e. purely neural or symbolic) influences performance. Overall, NEXUS enables interpretable and robust control via neuro-symbolic reasoning over high-level skills.

## 1 INTRODUCTION

Recent advancements in Deep Reinforcement Learning have led to highly capable agents on a diverse set of tasks (Mnih et al., 2015; Schulman et al., 2017; Gallici et al., 2024); however, these policies are most often based on neural networks that operate as black boxes and thus exhibit difficult to interpret behaviors that may be misaligned (Rudin, 2019). Without interpretability, identifying misalignment or correcting undesirable behaviors, remains a significant challenge for practitioners (Zahavy et al., 2016; Zhang et al., 2018; Delfosse et al., 2024b).

Neuro-symbolic approaches address this issue by combining neural networks for perceptual grounding with symbolic reasoning modules for decision-making (Delfosse et al., 2023a; Hazra & Raedt, 2023; Acharya et al., 2024; Delfosse et al., 2024b; Luo et al., 2024). These architectures aim to enhance transparency by representing the policies through symbolic structures that are more readily interpretable. Despite their promise, applying symbolic reasoning directly to low-level action spaces often results in policies of prohibitive complexity (cf. Figure 1, Section B). The combinatorial explosion of symbolic rules at fine-grained action levels undermines interpretability and scalability.

Hierarchical Reinforcement Learning offers an alternative by abstracting sequences of actions into higher-level skills or options (Sutton et al., 1999; Dietterich, 2000; Barto & Mahadevan, 2003). While the framework provides a natural structure, the resulting options, when learned autonomously, are often entangled or have overlapping goals, and the meta-policy remains opaque, both hindering interpretability.

To address these limitations, we draw inspiration from the dual-process theory of cognition (Kahneman, 2011), distinguishing fast, intuitive actions (System 1) from slow, deliberative reasoning (System 2). We emulate this cognitive structure in our hierarchical framework NEXUS, **N**eural **EX**ecution **U**nder **S**ymbols, which preserves the effectiveness of low-level neural policies (System 1), while combining them with a meta-policy layer of simple, interpretable rule sets (System 2) to maintain clarity about the reasoning behind the active skill in complex environments (cf. Figure 2). We ensure disentangled skills, i.e. each skill corresponds to a distinct situation with clear semantics, by defining their subgoals explicitly through reward functions on objects extracted from the image.

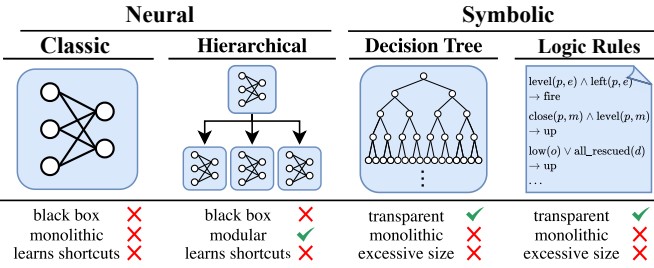

|  | **Neural** |  | **Symbolic** |  |
|---|---|---|---|---|
| **Classic** | **Hierarchical** |  | **Decision Tree** | **Logic Rules** |

| black box | ✗ | black box | ✗ | transparent | ✓ | transparent | ✓ |
| monolithic | ✗ | modular | ✓ | monolithic | ✗ | monolithic | ✗ |
| learns shortcuts | ✗ | learns shortcuts | ✗ | excessive size | ✗ | excessive size | ✗ |

Figure 1: **Current RL policies are not interpretable.** Neural network-based policies are non-modular black box models. Hierarchical RL separates high-level skill selection and low-level control, but the learned entangled skills. Transparent symbolic approaches leverage logical rules or decision trees on symbolic states, but quickly become excessively complex if applied on low-level state features and actions.

To reduce the manual effort required, we leverage the reasoning capabilities of Large Language Models (LLMs) to assist in identifying those skills, defining the high-level meta-policy functions, and generating the reward signals that guide the training of the low-level neural controllers. Both the low-level skills and the meta-policy Q-function are learned jointly in an off-policy manner. Overall, our neuro-symbolic design enables true interpretability on the abstract level of skills without compromising the efficiency of neural agents.

Our primary contributions are as follows:

**(i)** We extend *Parallelised Q-Networks* to the hierarchical setting for efficient and scalable meta-policy and skill learning (Section 3.1).

**(ii)** We introduce 3 NEXUS variants balancing interpretability and flexibility (Section 3.1 - Section 3.3).

**(iii)** We demonstrate that generated object-centric rewards and high-level meta-policy functions guide training towards disentangled skills and interpretable policies (Section 4.2).

**(iv)** We provide evidence that NEXUS agents are less susceptible to reward hacking and generalize better to small distribution shift than common algorithms (Section 4.3).

## 2 BACKGROUND

Let us introduce *Deep Reinforcement Learning*, that enables applying neural networks to sequential decision-making tasks. and *Hierarchical Reinforcement Learning*, that decomposes complex tasks into hierarchies of simpler sub-tasks, thereby abstracting actions into skills.

**Deep Reinforcement Learning.** Reinforcement Learning (RL) is a framework for sequential decision making in which an agent learns to interact with an environment in order to maximize cumulative reward. The environment is typically modeled as a Markov Decision Process (MDP), defined by the tuple $\mathcal{M} = \langle \mathcal{S}, \mathcal{A}, P, R, \gamma \rangle$, where $\mathcal{S}$ is the set of states, $\mathcal{A}$ is the set of actions, $P(s' \mid s, a)$ is the transition probability from state $s$ to state $s'$ under action $a$, $R(s, a)$ is the reward function and $\gamma \in [0, 1)$ is the discount factor. The goal of the agent is to learn a policy $\pi : \mathcal{S} \rightarrow \mathcal{A}$ that maximizes the expected discounted return: $\mathbb{E}_\pi \left[ \sum_{t=0}^\infty \gamma^t R(s_t, a_t) \right]$.

In *Q-learning*, a value-based RL algorithm, the agent seeks to learn the optimal action-value function $Q^*(s, a)$, which satisfies the Bellman optimality equation: $Q^*(s, a) = \mathbb{E}_{s'} \left[ R(s, a) + \gamma \max_{a'} Q^*(s', a') \right]$. This function is updated iteratively via the Q-learning update rule: $Q(s_t, a_t) \leftarrow Q(s_t, a_t) + \alpha \left[ r_t + \gamma \max_{a'} Q(s_{t+1}, a') - Q(s_t, a_t) \right]$, where $\alpha$ is the learning rate. With growing state space $\mathcal{S}$, it becomes infeasible to store and update a tabular $Q$-function. *Deep Q-Networks (DQN)* (Mnih et al., 2015) address this by using a deep neural network $Q_\theta(s, a)$, parameterized by $\theta$, to approximate $Q(s, a)$. DQN introduces several key modifications to stabilize learning, including experience replay, where transitions $(s_t, a_t, r_t, s_{t+1})$ are stored in a replay buffer and sampled randomly to break correlations between consecutive updates and target networks, where a separate network $Q_{\theta^-}(s, a)$ is used to compute the target value and is updated periodically: $y_t = r_t + \gamma \max_{a'} Q_{\theta^-}(s_{t+1}, a')$.

More recently, Gallici et al. (2024) introduced *Parallelised Q-Networks (PQN)*, a simplified variant of DQN that eliminates the use of experience replay and target networks. Instead, PQN leverages a large number of parallel (ideally vectorized) environments and applies normalization techniques to mitigate training instabilities. This high degree of parallelization enables substantially faster training while maintaining performance comparable to state-of-the-art RL algorithms.

Figure 2: **NEXUS with a neuro-symbolic meta-policy.** The symbolic state is extracted from the observation. The LLM-generated meta policy selects potential relevant skills, then selected using a trained meta Q-function. Finally, the selected neural skill outputs the final low-level action. While the meta policy's Q-function is trained using the environmental reward, each skill is trained using its own LLM-generated reward signal, allowing for interpretable disentangled skills.

**Hierarchical Reinforcement Learning.** Hierarchical reinforcement learning (HRL) decomposes complex tasks into simpler sub-tasks, often modeled as temporally extended actions or "options" (Sutton et al., 1999). At inference time, a meta policy selects the option, enabling more efficient exploration and transfer in environments with long horizons or sparse rewards. Sub-policies can be learned autonomously using intrinsic objectives (Bacon et al., 2017; Vezhnevets et al., 2017) or guided by manually designed reward functions (Sutton et al., 1999; Dietterich, 2000). However, autonomous discovery often suffers from option entanglement or collapse. NEXUS instead enforces disentangled options via explicit reward functions, which are LLM-generated.

## 3 Neural Execution Under Symbols

We address the challenge of building interpretable and high performing reinforcement learning agents with NEXUS, a hierarchical framework that combines symbolic reasoning with neural skill execution. The overall pipeline is described in Figure 2. At a high level, NEXUS decomposes decision-making into two layers: (1) a high-level meta-policy and (2) a set of low-level neural skill policies optimized for specific sub-goals. At each step, the decision process proceeds as follows. First, object-centric representations are extracted from pixel-based inputs. Second, symbolic rules, encoding the activation conditions of each skill, filter out skills without valid preconditions. Third, the learned meta-policy Q-function selects the most suitable skill among the remaining candidates. Finally, the chosen skill executes its policy to maximize the corresponding sub-goal. Object-centric representations are crucial for enabling symbolic rule definition and skill-specific reward design, both of which are difficult without such structured representations. Moreover, the explicit availability of object information allows the use of LLMs for generating the rules and reward functions.

In the following, we describe three distinct meta-policy function type: purely neural, purely symbolic, and neuro-symbolic. The first method, NEXUS (neural), corresponds to a hierarchical PQN method, which learns both a neural meta-policy and neural skills (based on skill-specific rewards) and which serves as the foundation for subsequent variants. While it offers high flexibility due to the learned meta-policy, it is missing reasoning for why a given sub-goal (and its corresponding skill) is selected in the current step. This shortcoming can be facilitated by replacing the learned meta-policy with a fixed, interpretable function that can be either manually defined or LLM-generated. This variant, NEXUS (symbolic), is fully symbolic at the meta-policy level, prioritizing transparency over adaptability. Finally, we present the hybrid approach NEXUS (nesy) that filters candidate skills via predefined symbolic rules and selects among them using Q-learning. This last variant results in a neuro-symbolic meta-policy, symbolic in rule-based filtering and neural in value estimation, combined with neural sub-policies, achieving a principled trade-off between interpretability and flexibility.

### 3.1 Hierarchical PQN

We accommodate the hierarchical learning setup by modeling the environment as two layers of MDPs. On the action level, we define a collection of MDPs $\mathcal{M} = \{\langle \mathcal{S}, \mathcal{A}, P, R_n, \gamma \rangle\}_{n=1}^N$ for $N$ skills, each associated with a distinct option policy $\pi_n \in \Pi$ that decides over the actual environment actions. The meta-level MDP $\mathcal{M}_{\text{meta}} = \langle \mathcal{S}, \Pi, P, R_{\text{env}}, \gamma \rangle$ governs option selection through a meta-policy $\pi_{\text{meta}}$ that selects over the options $\Pi$. Note that the option-level MDPs differ only in their reward functions $R_n$, while the meta-level MDP retains the environment's original reward $R_{\text{env}}$.

To enable learning across sub-policies even when they are not actively selected, we adopt off-policy Q-learning. This allows all the skills to learn from trajectories generated by the active skill and meta-policy while optimizing for their respective reward structures. Exploration within each skill is conducted via $\epsilon$-greedy action selection. For scalability through vectorization, we build our hierarchical training approach upon PQL with $\lambda$-returns (Gallici et al., 2024). However, instead of learning a single shared Q-network $Q_\phi$ as in PQL, we learn separate Q-functions $Q_{\phi_n}$ for each low-level skill, along with a meta-level Q-network $Q_{\phi_{\text{meta}}}$. Each Q-network thus predicts the returns of its own reward function.

For updating, we roll out trajectories $(s_0, ..., s_T)$ following the global policy by selecting the next active skill $\pi_n \in \Pi$ using the meta-policy Q-function $Q_{\phi_{\text{meta}}}$ and find the most promising action according to the active skill's Q-function $Q_{\phi_n}$:

$$\pi_n = \arg\max_{\pi_n \in \Pi} Q_{\phi_{\text{meta}}}(s_t, \pi_n), \qquad a' = \arg\max_{a' \in \mathcal{A}} Q_{\phi_n}(s_t, a'). \tag{1}$$

By executing the action in the environment, we obtain the rewards and the next environmental state for the next iteration. We employ $\epsilon$-greedy for exploration during both the skill and the action selection.

Next to the environment reward $r_{\text{env},t}$, we require skill-specific reward functions $r_{n,t}$ that are based on the object-centric state $s_t$ and may be automatically generated by LLMs (cf. Section G), which we use to compute the $\lambda$-returns recursively back in time (for details, we refer the reader to Gallici et al. (2024); Daley & Amato (2019)):

$$R_{n,t}^\lambda = r_{n,t} + \gamma \left[ \lambda R_{n,t+1}^\lambda + (1 - \lambda) \max_{a'} Q_{\phi_n}(s_{t+1}, a') \right], \tag{2}$$

and similarly, for the learned meta-policy using environment rewards:

$$R_{\text{env},t}^\lambda = r_{\text{env},t} + \gamma \left[ \lambda R_{\text{env},t+1}^\lambda + (1 - \lambda) \max_{\pi_n} Q_{\phi_{\text{meta}}}(s_{t+1}, \pi_n) \right], \tag{3}$$

or, if $s_t$ is terminal $R_{n,t}^\lambda = r_{n,t}$ and $R_{\text{env},t}^\lambda = r_{\text{env},t}$. All learned Q-functions are updated towards their $\lambda$-returns. We provide the full algorithm in Section C.

## 3.2 INTERPRETABLE META-POLICY FUNCTION

While the hierarchical structure allows to identify the active skill, it does not expose the decision process that leads to its selection. To make this process transparent, we introduce interpretable meta-policy functions implemented as rule-based programs. We choose this representation for two key reasons. First, rule-based programs offer simplicity and accessibility since conditional statements can typically be understood and modified even by users with limited programming background. Second, LLMs are highly effective at generating and editing code-like structures due to extensive pretraining on programming data. This enables generation of human-interpretable meta-policies and greatly reduces manual effort. Further details on how we employ LLMs to generate the meta-policy function are available in Section G.

Formally, instead of relying on a learned Q-function $Q_{\phi_{\text{meta}}}$, we define a meta-policy $\pi_{\text{meta}} : \mathcal{S} \to \Pi$ as a set of human-readable rules that directly maps the current object-centric state $s_t \in \mathcal{S}$ to the selected low-level policy $\pi_n \in \Pi$:

$$\pi_n = \pi_{\text{meta}}(s_t), \tag{4}$$

We hereby constrain $\pi_{\text{meta}}$ to simple rules that are mutually exclusive to enable transparent inspection of policy decisions. The selected skill can be traced back to the specific rule that evaluates to true given the current state (cf. Figure 5, left side).

## 3.3 NEURO-SYMBOLIC META-POLICY FUNCTION

The symbolic meta-policy is a deterministic function that selects skills based on the rule with the highest priority. However, this approach may become inefficient in situations where multiple skills are simultaneously applicable. Consider an agent controlling a submarine that can shoot a nearby enemy or surface to replenish oxygen. The most viable depends on the specific context: if the oxygen level is critically low, resurfacing may be prioritized; if an enemy is dangerously close, attacking may take precedence. Crafting hard-coded rules to handle such trade-offs would quickly lead to growing, less interpretable policies.

Instead, we propose a neuro-symbolic (NeSy) approach, by maintaining a set of high-level, interpretable conditions (e.g., "go to surface if oxygen is low", "fight if enemy is close"), represented as a binary condition vector $c_t \in \{0, 1\}^N$, where each entry indicates whether a particular condition is active at time $t$. We formulate these rule-sets again as simple code functions and leverage LLMs for their generation (cf. Figure 5, right side). However, this time we allow multiple conditions to be active concurrently. The skill selection is then modulated by these conditions using a binary mask applied to the meta-policy Q-values:

$$\pi_n = \arg\max_{\pi_n \in \Pi} \left( c_t \odot Q_{\phi_{\text{meta}}}(s_t, \pi_n) \right) \tag{5}$$

where $\odot$ denotes element-wise multiplication, $Q_{\phi_{\text{meta}}}(s_t, \cdot) \in \mathbb{R}^N$ is the vector

---

**Algorithm 1** NEXUS Variants

**Require:** Skill policies $\{\pi_n\}_{n=1}^N$ with Q-functions $\{Q_{\phi_n}\}_{n=1}^N$, meta-policy $Q_{\phi_{\text{meta}}}$ (for learned/soft)
1: **for** each episode **do**
2:   **for** each environment step (in parallel) **do**
3:     Extract object-centric state $s_t$
4:     Select skill $\pi_{n,t}$ as follows:
     A) Neural: $\pi_{n,t} = \arg\max_{\pi_n} Q_{\phi_{\text{meta}}}(s_t, \pi_n)$
     B) Symbolic: $\pi_{n,t} = \pi_{\text{meta}}(s_t)$
     C) NeSy: $\pi_{n,t} = \arg\max_{\pi_n} (c_t \odot Q_{\phi_{\text{meta}}}(s_t, \pi_n))$
5:     Select action $a_t$ using $\epsilon$-greedy from $\pi_{n,t}$
6:     Execute $a_t$ and observe transition $(\{r_{n,t}\}_{n=0}^N, r_{\text{env},t}, s_{t+1})$
7:   **end for**
8:   **for** each gradient step **do**
9:     Compute $R_{n,t}^\lambda$ and $R_{\text{env},t}^\lambda$ (if A or C)
10:     Update each $Q_{\phi_n}$ using skill-specific $R_{n,t}^\lambda$
11:     Update $Q_{\phi_{\text{meta}}}$ using $R_{\text{env},t}^\lambda$ (if A or C)
12:   **end for**
13: **end for**

---

of meta-policy Q-values for each interpretable condition, and $\Pi$ is the set of available skills.

This way, we can preserve interpretability at the symbolic level while enabling the agent to resolve ambiguous scenarios adaptively based on learned preferences. An overview of the different meta-policy variations is provided in Algorithm 1.

**Prior knowledge requirements.** There are two components of NEXUS that require prior knowledge. The first is the specification of skill reward functions. The second is the definition of meta-policy rules for the symbolic and the neuro-symbolic variants. Although these elements can be provided manually, we leverage LLMs to minimize expert input. Given the prompts in Section G, only the game manual is ultimately required.

## 4 EXPERIMENTAL EVALUATION

This work aims to develop an object-centric pipeline for agents capable of solving complex environments while maintaining interpretability of high-level goals. We assess this objective by addressing the following research questions:

**(Q1)** Does NEXUS learn meaningful disentangled skills?
**(Q2)** Are NEXUS policies interpretable?
**(Q3)** Can NEXUS compete with other deep methods?
**(Q4)** Does NEXUS improve robustness to game simplifications?
**(Q5)** How does noise in the object extractor influence NEXUS?

### 4.1 EXPERIMENTAL SETUP

Our experiments assume access to pre-extracted object representations and attributes. We therefore provide the agent with symbolic environment states directly, allowing us to remain agnostic to the specific object extraction pipeline. We conduct experiments on *JAXAtari*[1], a JAX-based reimplementation of the Atari Learning Environment (ALE) (Bellemare et al., 2013), specifically using the games *Kangaroo* and *Seaquest*, with object-centric representations similar to *OCatari* (Delfosse et al., 2024a). These games are selected because their gameplay naturally decomposes into low-level skills that must be combined to solve the overall task. In Kangaroo, the objective is to ascend through platforms while avoiding enemies and collecting berries. In Seaquest, the player navigates a submarine to rescue divers and return them to the surface, while avoiding hazards such as sharks

---

[1]https://github.com/k4ntz/JAXAtari

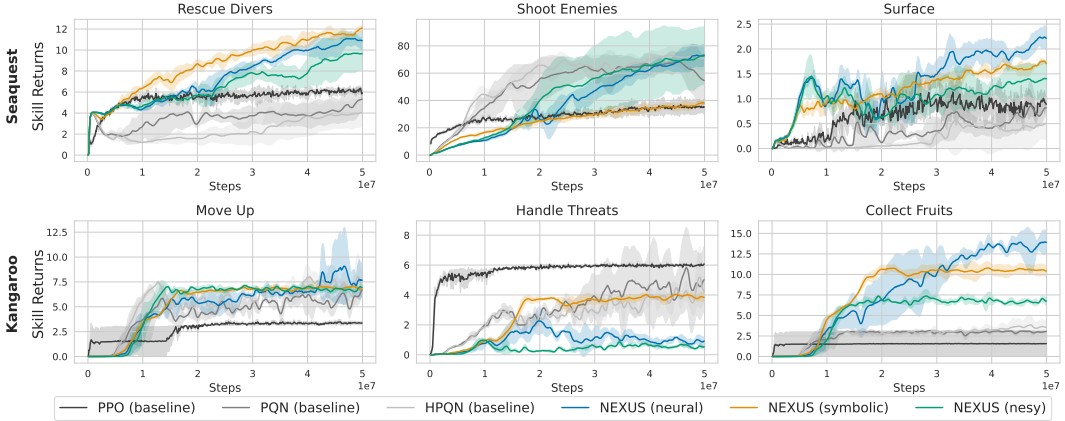

Figure 3: **NEXUS learns disentangled skills from off-policy data.** In Seaquest (top), the baseline methods mainly focus on shooting enemies, while the NEXUS approaches acquire the target skills more evenly. Similarly, in Kangaroo (bottom), the NEXUS approaches learn 'Move Up' and 'Collect Fruits' reliably, while the baselines focus mostly on 'Handle Threats'.

and enemy submarines. These two games are known to exhibit *reward hacking* behavior, meaning that the agents find simple shortcuts of increasing their return rather than the intended way that humans would typically choose. Examples include exclusively shooting enemies in Seaquest and only catching falling apples or boxing monkeys in Kangaroo. Additionally, we evaluate on *Crafter* (Hafner, 2021), using the JAX reimplementation by Matthews et al. (2024), an environment that integrates elements from the games Minecraft and NetHack. Unlike Atari games, Crafter presents a more complex, open-ended challenge that requires agents to acquire and coordinate multiple skills, including navigation, combat, resource collection, and crafting. While primarily designed as a benchmark for open-ended learning, Crafter's task diversity makes it well-suited for evaluating the generalization and skill composition capabilities of RL methods.

Finally, for generalization we also test the pre-trained agents on slightly modified versions of the games. These modifications are designed to highlight the sensitivity of standard reinforcement learning agents to minor changes in the environment, including simplifications. For example, removing enemies from Seaquest to reduce the task's complexity already leads to a substantial drop in many deep agents performances (Delfosse et al., 2025).

We compare all three variations of NEXUS to the default, non-interpretable PQN (Gallici et al., 2024), and, in case of the Atari games, with the actor-critic PPO (Schulman et al., 2017) as baselines. Additionally, we compare to HPQN, a hierarchical PQN variation that does not employ skill-specific rewards and has a purely neural meta-policy. All results (including baselines) are directly trained on object-centric inputs, which may affect performance compared to image based training. We adhere to the standard frame budgets of both games: 200M frames for JAXAtari and 1B frames for Crafter. Further implementation details are available in Section D.

## 4.2 INTERPRETABILITY RESULTS

**Disentangled skill learning (Q1).** We first evaluate whether NEXUS enables efficient learning of meaningful skills. Since each skill requires its own reward function, we leverage the reasoning capabilities of LLMs to generate them based on the game manual, skill definitions, and the object-centric state (cf. Section G for further information).

Figure 3 demonstrates that NEXUS is able to learn most target skills successfully from a single source of off-policy data. This means that a skill can learn from another skills action and is not required to be activated to do so. Unlike the baselines, which tend to mostly focus on a single skill that maximizes environment reward (e.g., Shoot Enemies for Seaquest, Handle Threats for Kangaroo), NEXUS approaches promote balanced skill acquisition. Especially the symbolic meta-policy approach seems to learn all skills most reliably.

Figure 4: **NEXUS succesfully learns LLM-proposed Crafter skills.** Compared to the baseline, NEXUS approaches often converge faster on the proposed skills.

```python
def meta_policy(st: state):
    if enemy_close(st.enemies,
    st.player):
        return fight_enemies()
    elif is_available(st.divers):
        return rescue_divers()
    elif is_low(st.oxygen):
        return surface()
    elif all_collected(st.divers):
        return surface()
    return rescue_divers()
```

```python
def meta_policy_rules(st: state):
    fight_enemies = False
    rescue_divers = True
    surface = False
    if enemy_close(st.enemies,
    st.player):
        fight_enemies = True
    if is_low(st.oxygen) or
    all_collected(st.divers):
        surface = True
    return [fight_enemies,
    rescue_divers, surface]
```

Figure 5: **NEXUS policies and rules are clear and interpretable.** A symbolic meta-policy (left) and similar filtering rules for the neuro-symbolic meta-policy (right) for the Atari game Seaquest.

We experience a similar picture when evaluating on Crafter. Using the game manual and state description, we query an LLM for essential skills. Recurrent versions of NEXUS are then trained explicitly for these skills and compared to a recurrent PQN baseline. Results are presented in Figure 4, where we compare the ability to learn the skills between the different methods. Similar to before, NEXUS approaches are able to learn the five skills with symbolic being the fastest.

**Interpretability of NEXUS policies (Q2).** To assess the interpretability of NEXUS, we visualize a fixed meta-policy for Seaquest on the left side in Figure 5. By abstracting raw observations into object-centric representations and low-level actions into high-level skills, the decision-making process becomes transparent. Each option has a clear, mutually exclusive activation condition, enabling unambiguous skill selection. For example, the combat skill activates if and only if an enemy is in close proximity. This simplicity offers two advantages: technical users can design fully interpretable policies, and LLMs can autonomously generate such meta-policies, which remain editable due to their transparency. We illustrate LLM-based meta-policy generation in Section G.

Full transparency is not guaranteed for neuro-symbolic meta-policies, as multiple conditions may be simultaneously satisfied (e.g., an enemy is nearby and oxygen is low; cf. Figure 5, right side). Nonetheless, such overlaps are rare, and interpretability is largely preserved. In ambiguous cases, interpretability is traded for flexibility, allowing the agent to select the skill with the highest expected return based on the meta-policy Q-values. An example of a Seaquest agent operating under a neuro-symbolic meta-policy is available in Figure 6.

### 4.3 PERFORMANCE RESULTS

**Comparison to other approaches (Q3).** We evaluate NEXUS against baselines using two metrics: Game returns and aligned environment goals, which track progress toward goals defined in game manuals (e.g. , divers rescued in Seaquest, level completion in Kangaroo). The latter aims to capture the overall alignment with the intended game objectives, which differ from just maximizing the reward for these environments, where deep agents usually perform reward hacking (Shihab et al., 2025).

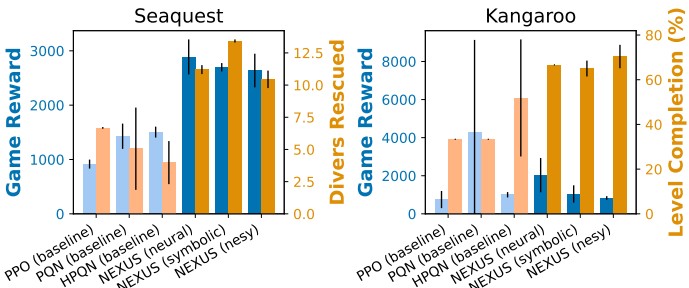

Figure 6: **NEXUS produces interpretable yet flexible high-level plans for ambiguous scenarios.**
Left: Both "enemy_closeby" and "diver_available" rules evaluate to true; the learned meta-policy
prioritizes fighting, likely due to the diver's proximity to the enemy. Middle: Under similar conditions,
rescuing the diver is preferred, reason could be the easier access. Right: With all divers collected and
an enemy nearby, the meta-policy opts to return to the surface due to the higher estimated return.

As shown in Figure 7, NEXUS
approaches are competitive to
the baselines. In most cases,
they achieve comparable or even
higher HNS and notably outper-
form the baselines on the actual
games' main goal, demonstrat-
ing the baselines reward hack-
ing tendency. The results sug-
gest that NEXUS mitigates re-
ward misalignment by incorpo-
rating domain priors into the
decision-making process, while
still achieving good performance.
We provide further comparisons
to neuro-symbolic and interpretable RL methods in Section E.

Figure 7: **NEXUS is competitive.** In Seaquest, NEXUS out-
performs the baselines on both HNS and rescuing divers. In
Kangaroo, NEXUS approaches are better aligned to the actual
game objective.

**Robustness to game simplifications (Q4).** Most RL algorithms struggle to adapt to even minor
variations in the environment (Delfosse et al., 2025). Surprisingly, their performances drop even in
settings that simplify the game for humans, such as removing deadly threats like enemies and their
projectiles in Seaquest and Kangaroo. We assess whether NEXUS agents can generalize to such
simplified variants by training the agents on the standard versions of the games and evaluating on the
unseen simplifications. In Seaquest and Kangaroo we remove all threats, while in Crafter we remove
the need to drink water for survival. Figure 8 presents the results. As expected, the baselines suffer
substantial performance degradation under the simplifications in all three games.

All NEXUS variants exhibit distinct robustness characteristics. In Atari, only the symbolic variant
demonstrates improved robustness, with smaller performance drops in Kangaroo and even gains in
Seaquest, while the fully neural and neuro-symbolic variants show limited robustness. In Crafter all
NEXUS variants maintain or improve performance. We attribute this robustness to symbolic steering
of the meta-policy, which deactivates unnecessary skills such as handling enemies or drinking water.

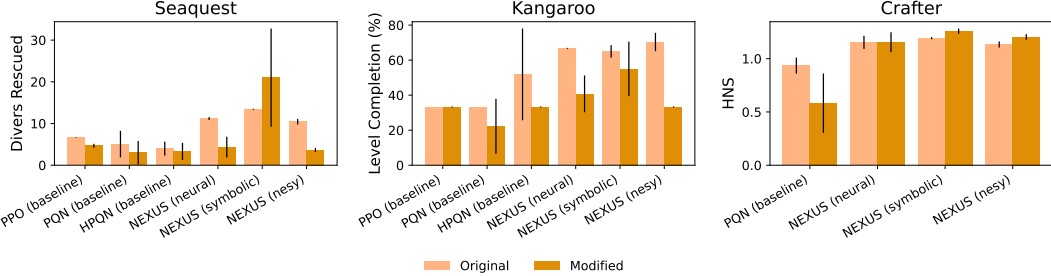

Figure 8: **Symbolic NEXUS remains practical on simplified games.** While baseline performance
drops significantly, neuro-symbolic and fully symbolic NEXUS improve in the modified Seaquest
(left) and symbolic NEXUS shows smaller performance drops in modified Kangaroo (middle).

**Influence of noisy detections on NEXUS (Q5).** While object detection methods become increasingly reliable, misdetections still happen. With the following ablation, we test whether NEXUS is robust to misdetections and noise in the detections. For that, we incorporate a 10% misdetection chance and add gaussian noise with a standard deviation of 3px to each object attribute during training. The results are visualized in Figure 9. We observe that both the neural and the symbolic approaches

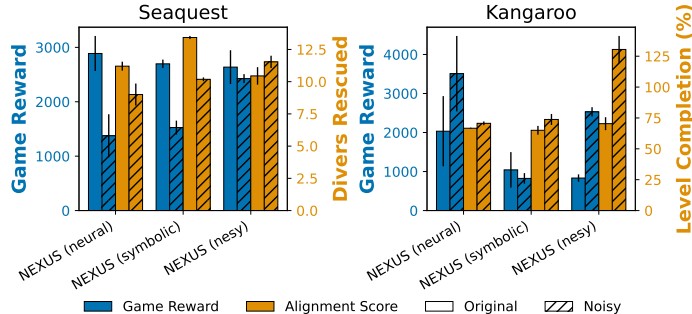

Figure 9: **Effect of noisy detections.** In Seaquest, the neural and the symbolic meta-policy take a performance hit in both reward and alignment score, while the NeSy variation is robust. In Kangaroo, all approaches improve with the added noise.

lose performance when the noise is applied in Seaquest, while the neuro-symbolic meta-policy remains reliable. Surprisingly, the experiments on Kangaroo indicate that all approaches increase the level completion score and often times the game reward. Most notably, the neuro-symbolic meta-policy is able to increase the level completion rate from ∼70% to above 125% and always finishes level one. Further details and results are available in Section F.

**On the choice of the meta-policy.** Drawing from our findings, we can now offer recommendations regarding the optimal design of the meta-policy. The purely neural meta-policy presents the simplest training paradigm as it does not need a separate symbolic meta-policy, and it demonstrates strong performance in terms of training environment reward. However, this approach sacrifices interpretability and also performance when evaluated on simplified environments. In contrast, the purely symbolic meta-policy necessitates an additional step of rule definition. Since this can be largely automated with LLMs, this investment is often justified by its enhanced interpretability and robustness. Lastly, the neuro-symbolic approach eases the definition of the additional rules, since they do not need to be mutually exclusive. While the performance is often similar to the symbolic policy, it is less robust to game simplification, but more robust to noise in the detection method. Considering these trade-offs, we advocate for both, the purely symbolic and the neuro-symbolic meta-policies as effective choices, offering a compelling balance of strong performance, interpretability, and generalizability.

**Limitations.** NEXUS relies on object-centric scene decoders that accurately provide the agent with objects and their positions and sizes from raw images, which we assume to exist in this work. For Atari games, multiple approaches are viable (Li et al., 2017; Locatello et al., 2020; Lin et al., 2020; Delfosse et al., 2023b) with some achieving near-perfect sprite extraction (Smirnov et al., 2021). For real-world data, recent advancements have significantly improved robustness, with models like SAM2 (Ravi et al., 2025) reaching up to 90% $\mathcal{J}\&\mathcal{F}$ accuracy on zero-shot segmentation.

NEXUS requires a pre-defined description of the task, i.e. the game's manual, for the reward generation (as done in Wu et al. (2023)). This limits it's applicability to tasks that can be explained in language. Moreover, while the LLM-generated reward functions and meta-policies generally capture valid semantics and are logically consistent, some manual adjustments to align them with the implementation framework are still necessary. Prior work on LLM-based reward design (e.g., Xie et al. (2024), Ma et al. (2024), Kaufmann et al. (2024)) has documented recurring issues such as reward misspecification, proxy objective selection, and over- or under-constrained preconditions, highlighting the need for careful verification. NEXUS currently addresses these risks through manual inspection of the generated code, but does not incorporate automated diagnostics or iterative improvements during learning. Extending our framework by incorporating more refined mechanisms is a crucial direction for enhancing robustness and scalability. Lastly, the presented approach is based on Q-learning and thus currently limited to discrete action space, however, extension to continuous action spaces seems viable by adopting an off-policy actor critic instead of $\epsilon$-greedy action selection (Lillicrap et al., 2016; Haarnoja et al., 2018).

## 5   RELATED WORK

**Interpretable and Hierarchical Reinforcement Learning.** Interpretability in RL can be introduced at various stages of the pipeline (Glanois et al., 2024), often by deriving symbolic state representations from raw observations via object recognition or segmentation (Li et al., 2017; Locatello et al., 2020; Kirillov et al., 2023; Lin et al., 2020; Delfosse et al., 2023b). Such object-centric states have enabled interpretable policies through decision trees (Silva et al., 2019; Likmeta et al., 2020; Delfosse et al., 2024b), logic rules (Maes et al., 2012; Akrour et al., 2019; Delfosse et al., 2023a), parametric functions (Luo et al., 2024) or programmatic policies and trees (Verma et al., 2019; Anderson et al., 2020; Kohler et al., 2024). Neural and logical policies can also be efficiently combined (Shindo et al., 2025). Complementary efforts introduce hierarchical decompositions, where high-level interpretable policies select among low-level sub-policies, leveraging annotated task sketches (Andreas et al., 2017) or (differentiable) symbolic planning (Leonetti et al., 2016; Yang et al., 2018; Jin et al., 2022; Lyu et al., 2019; Ye et al., 2025). Hierarchical symbolic planning based on object-centric representations has been shown to be beneficial for task transferability (James et al., 2022) and robotics applications (Sharma et al., 2020). Our work extends these directions by enforcing semantic separation of skills, introducing a neurosymbolic meta-policy to balance interpretability and flexibility, and integrating LLMs throughout the pipeline. Unlike prior approaches, we validate on challenging Atari and Crafter environments.

**Relation to the Options framework.** HRL has been studied extensively, with the Options framework (Sutton et al., 1999) as the most prominent formulation. NEXUS instantiates the Options framework by performing intra-option learning, where multiple options are learned simultaneously from shared off-policy experience, using sub-policies guided by option-specific reward functions and coordinated by a neuro-symbolic meta-policy. Key differences arise in temporal abstraction and in the treatment of initiation and termination conditions. Rather than executing an option until termination, the meta-policy selects the active option at every time step, jointly determining activation and termination. The logical rule set used to filter Q-values before selection serves as a generalized initiation set, enabled by the object-centric encoding of observations. NEXUS' key innovations to HRL are: (1) disentangled neural options through specialized rewards and (2) interpretable meta-policies.

## 6   CONCLUSION

In this work, we present NEXUS, a hierarchical RL method that combines high interpretability on meta-policy level with neural, low-level action execution. The evaluation suggest several advantages of our approach. It learns disentangled sub-policies corresponding to identifiable skills, provides interpretable and modular structures for inspection and intervention, reduces reward-hacking through fine-grained control, and remains robust to environment simplifications where standard deep RL agents fail. We also demonstrate that LLMs can be integrated into the NEXUS pipeline to generate skill decompositions, reward functions, and symbolic meta-policies, enabling dynamic adaptation to novel objects, evolving environments, or shifting task objectives. Future work should increase the autonomy of policy adaptation by incorporating mechanisms to detect when new skills or meta-policy updates are required, e.g. via causal world models (Yang et al., 2025; Dillies et al., 2025), which could enable scaling to complex, open-ended environments. Updates to the skills or meta-policy could be retrieved by re-querying an LLM. Additionally, future work should evaluate the actual interpretability of NEXUS in a user-study. Overall, this work advances RL interpretability and modularity through symbolic and object-centric reasoning while supporting human-in-the-loop control at the skill level, offering a promising path toward transparent, adaptable, and aligned agents.

**Reproducibility Statement.**   We have taken several measures to ensure reproducibility of our results. Details of the proposed method, including model architectures, training procedures, evaluation protocols, hyperparameters, implementation details and LLM interactions are provided in the main paper and the appendix. Additionally, we release source code and configuration files to reproduce all experiments, along with environment setup instructions.

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

## A  LLM USAGE

Beyond integration within the pipeline (cf. Section G), LLMs were used solely to improve text readability and generate boilerplate code.

## B  MOTIVATION - INTERPRETABLE POLICY SIZES

We visualize the actual logic policies retrieved from existing interpretable RL methods in Listing 1 and Listing 2. While these policies have maximum transparency, they are difficult to interpret for humans due to the massive size. NEXUS on the other hand presents a simple solution by incorporating hierarchical abstraction and thus allowing for compact and truly interpretable policies, cf. Figure 5.

```python
def play ( state ):
    if state .Ball_0.prev_x <= 1.24:
        if state .Ball_0.y - state .Enemy_0.prev_y <= -0.41:
            if state .Ball_0.x - state .Ball_0.prev_x <= 0.09:
                if state .Player_0.y - state .Enemy_0.y <= -0.91:
                    if state .Ball_0.x <= 0.73:
                        if state .Player_0.y - state .Ball_0.x <= -1.20:
                            return "LEFT"
                        else :
                            return "RIGHT"
                    else :
                        return "NOOP"
                else :
                    if state .Ball_0.x <= -0.05:
                        return "NOOP"
                    else :
                        if state .Player_0.y - state .Player_0.prev_y <= -0.19:
                            return "NOOP"
                        else :
                            if state .Ball_0.prev_x - state .Ball_0.prev_y <= -0.38:
                                return "NOOP"
                            else :
                                if state .Ball_0.x - state .Ball_0.prev_x <= -0.09:
                                    return "RIGHT"
                                else :
                                    return "RIGHT"
            else :
                return "RIGHT"
        else :
            if state .Ball_0.x - state .Enemy_0.y <= -1.46:
                if state .Player_0.y - state .Ball_0.prev_x <= -0.92:
                    return "LEFT"
                else :
                    return "RIGHT"
            else :
                if state .Player_0.y - state .Ball_0.y <= -0.63:
                    if state .Ball_0.prev_x <= -0.05:
                        return "LEFT"
                    else :
                        if state .Player_0.y - state .Ball_0.y <= -0.80:
                            return "LEFT"
                        else :
                            if state .Ball_0.prev_x <= 0.80:
                                return "LEFT"
                            else :
                                return "NOOP"
                else :
                    if state .Ball_0.x <= -0.07:
                        if state .Ball_0.prev_y <= -0.15:
                            if state .Ball_0.prev_y <= -1.96:
                                return "LEFT"
                            else :
                                return "LEFT"
                        else :
                            return "NOOP"
                    else :
                        if state .Ball_0.y <= 1.24:
                            if state .Player_0.y - state .Ball_0.y <= -0.48:
```

```python
                                if state.Ball_0.prev_x <= 0.68:
                                    return "NOOP"
                                else:
                                    return "NOOP"
                            else:
                                if state.Ball_0.x <= 0.54:
                                    if state.Player_0.y <= -0.82:
                                        return "NOOP"
                                    else:
                                        if state.Enemy_0.y - state.Enemy_0.prev_y <= -0.09:
                                            return "RIGHT"
                                        else:
                                            return "RIGHT"
                                else:
                                    if state.Ball_0.x - state.Ball_0.y <= -0.21:
                                        return "NOOP"
                                    else:
                                        return "RIGHT"
                        else:
                            if state.Ball_0.y <= 1.54:
                                return "NOOP"
                            else:
                                return "LEFT"
    else:
        if state.Player_0.y - state.Ball_0.y <= -0.54:
            if state.Player_0.y - state.Ball_0.y <= -0.88:
                return "LEFT"
            else:
                if state.Ball_0.y <= -1.40:
                    return "RIGHT"
                else:
                    return "NOOP"
        else:
            if state.Enemy_0.y - state.Enemy_0.prev_y <= 0.93:
                return "RIGHT"
            else:
                return "LEFT"
```

Listing 1: Pong policy of SCoBots (Delfosse et al., 2024b)

```
up_air(X):-oxygen_low(B).
up_divers_collected(X):- all_divers_collected(D).
fire_left(X):-same_depth_enemy(P,E),visible_enemy(E), facing_left(P),right_of_enemy(P,E).
fire_right(X):-same_depth_enemy(P,E),visible_enemy(E), facing_right(P),left_of_enemy(P,E).
left_aim(X):-right_of_enemy(P,E), facing_right(P),same_depth_enemy(P,E),visible_enemy(E).
right_aim(X):-left_of_enemy(P,E), facing_left(P),same_depth_enemy(P,E),visible_enemy(E).
down_aim(X):-higher_than_enemy(P,E),visible_enemy(E).
up_aim(X):-deeper_than_enemy(P,E),visible_enemy(E).
up_evade(X):-close_by_enemy(P,E),same_depth_enemy(P,E),visible_enemy(E).
down_evade(X):-close_by_enemy(P,E),same_depth_enemy(P,E),visible_enemy(E).
up_evade(X):-close_by_missile(P,M),same_depth_missile(P,M), visible_missile(M).
down_evade(X):-close_by_missile(P,M),same_depth_missile(P,M), visible_missile(M).
left_to_diver(X):- right_of_diver(P,D),close_by_diver(P,D), visible_diver(D).
right_to_diver(X):- left_of_diver(P,D),close_by_diver(P,D), visible_diver(D).
up_to_diver(X):-deeper_than_diver(P,D),close_by_diver(P,D), visible_diver(D).
down_to_diver(X):-higher_than_diver(P,D),close_by_diver(P,D), visible_diver(D).
```

Listing 2: Seaquest policy of NUDGE (Delfosse et al., 2023a)

```
logits_noop1  = −0.56∗y_agent_1∗∗2 − 0.38∗y_agent_1∗y_agent_2 − 0.087∗y_agent_1∗
     y_opponent_1 − 0.16∗y_agent_1∗y_opponent_2 − 0.76∗y_agent_1∗y_opponent_3 − 0.51∗
     y_agent_1∗y_opponent_4 − 0.54∗y_agent_1 − 0.24∗y_agent_2∗∗2 − 0.073∗y_agent_2 +
     0.27∗y_agent_4∗∗2 + 0.55∗y_agent_4 − 0.078∗y_opponent_1∗∗2 − 0.33∗y_opponent_1∗
     y_opponent_2 − 0.2∗y_opponent_1 − 0.35∗y_opponent_2∗∗2 − 0.5∗y_opponent_2 − 0.34∗
     y_opponent_3∗∗2 − 0.45∗y_opponent_3∗y_opponent_4 − 0.32∗y_opponent_3 − 0.15∗
     y_opponent_4∗∗2 − 0.19∗y_opponent_4 + 1.1
logits_noop2  = −0.074∗y_agent_1∗y_opponent_2 + 0.059∗y_agent_1∗y_opponent_3 − 0.097∗
     y_agent_4 − 0.16∗y_opponent_1∗y_opponent_2 − 0.18∗y_opponent_2∗∗2 − 0.27∗
     y_opponent_2 + 0.063∗y_opponent_4
logits_up1  = 0.23∗y_agent_1∗∗2 + 0.59∗y_agent_1∗y_agent_2 + 0.4∗y_agent_2∗∗2 + 0.11∗
     y_agent_2 − 1.5∗y_agent_4∗∗2 − 3.6∗y_agent_4 + 0.068∗y_opponent_3 + 1.1
logits_down1 = 0.09∗x_ball_3 + 0.12∗x_ball_4 − 0.21∗y_agent_1∗∗2 + 0.12∗y_agent_1∗
     y_opponent_1 + 0.27∗y_agent_1∗y_opponent_2 − 0.43∗y_agent_1∗y_opponent_3 − 0.28∗
     y_agent_1∗y_opponent_4 + 0.13∗y_agent_2 + 0.14∗y_agent_4∗∗2 + 0.43∗y_agent_4 +
     0.087∗y_ball_3 + 0.15∗y_ball_4 + 0.14∗y_opponent_1∗∗2 + 0.6∗y_opponent_1∗
     y_opponent_2 + 0.61∗y_opponent_1 + 0.65∗y_opponent_2∗∗2 + 1.1∗y_opponent_2 − 0.2∗
     y_opponent_3∗∗2 − 0.26∗y_opponent_3∗y_opponent_4 − 2.8∗y_opponent_3 − 0.085∗
     y_opponent_4∗∗2 − 0.14∗y_opponent_4 − 2.3
logits_up2  = 0.063∗x_ball_4 − 0.078∗y_agent_1 + 0.18∗y_agent_2∗∗2 + 0.52∗y_agent_2∗
     y_agent_3 + 0.35∗y_agent_2∗y_opponent_1 + 0.29∗y_agent_2∗y_opponent_2 + 0.26∗
     y_agent_2 + 0.38∗y_agent_3∗∗2 + 0.51∗y_agent_3∗y_opponent_1 + 0.42∗y_agent_3∗
     y_opponent_2 + 1.6∗y_agent_3 − 8.2∗y_agent_4 − 0.085∗y_ball_3 + 0.17∗y_opponent_1∗∗2
      + 0.28∗y_opponent_1∗y_opponent_2 + 0.25∗y_opponent_1 + 0.11∗y_opponent_2∗∗2 +
     0.15∗y_opponent_2 − 0.074∗y_opponent_3 + 0.26 logits_down2 = −0.052∗x_ball_1 − 0.068∗
     x_ball_3 − 0.093∗x_ball_4 + 0.18∗y_agent_1 − 0.17∗y_agent_2∗∗2 − 0.49∗y_agent_2∗
     y_agent_3 − 0.33∗y_agent_2∗y_opponent_1 − 0.27∗y_agent_2∗y_opponent_2 − 0.39∗
     y_agent_2 − 0.35∗y_agent_3∗∗2 − 0.48∗y_agent_3∗y_opponent_1 − 0.4∗y_agent_3∗
     y_opponent_2 − 0.38∗y_agent_3 + 0.15∗y_agent_4∗∗2 + 0.54∗y_agent_4 − 0.06∗y_ball_1 −
     0.064∗y_ball_3 − 0.11∗y_ball_4 − 0.17∗y_opponent_1∗∗2 − 0.28∗y_opponent_1∗
     y_opponent_2 − 0.58∗y_opponent_1 − 0.13∗y_opponent_2∗∗2 − 0.38∗y_opponent_2 + 2.2∗
     y_opponent_3 − 0.052∗y_opponent_4 − 3.6

action_noop  = [exp(logits_noop1) + exp(logits_noop2)]  /  sum(exp(logits))
action_up   = [exp(logits_up1)  + exp(logits_up2)]  /  sum(exp(logits))
action_down = [exp(logits_down1) + exp(logits_down2)]  /  sum(exp(logits))
```

Listing 3: Pong policy of INSIGHT (Luo et al., 2024)

## C  HIERARCHICAL PQN ALGORITHM

The complete algorithm for hierarchical PQN with a neural meta-policy is provided in Algorithm 2.

## D  IMPLEMENTATION DETAILS

**Setup.**   We base our implementations on CleanRL (Huang et al., 2022) and PureJaxRL (Lu et al., 2022), adapting them to object-centric inputs by replacing convolutional encoders with lightweight MLPs for feature extraction. Hyperparameters are listed in Table 1 and Table 2 and remain largely consistent with the original implementations, except for an increased number of parallel environments enabled by the efficiency of JAX-based code.

The exploration parameter $\epsilon$ was selected via a brief hyperparameter sweep in the range $[1, 0.001]$, using final test return as the selection criterion.

Each experiment is run with three random seeds (0, 1, 2) to ensure reproducibility. Reported plots include the corresponding standard deviation.

All experiments were conducted on a single NVIDIA Tesla V100-SXM3-32GB-H GPU on an NVIDIA DGX Server (Version 5.1.0) with CUDA 12.4.

**Algorithm 2** Hierarchical PQN

---

**Require:** Update period $U$, number of parallel environments $E$, number of skills $N$, exploration probability $\epsilon$

**Ensure:** Learned Q-network parameters $\{\phi_n\}_{n=1}^N$, $\phi_{\text{meta}}$

1: Initialize Q-network parameters $\{\phi_n\}_{n=1}^N$, $\phi_{\text{meta}}$
2: Sample initial states $s_0^e \sim P_0$ for $e \in \{0, \ldots, E-1\}$
3: $t \leftarrow 0$
4: **for** each episode **do**
5:     **for all** $e \in \{0, \ldots, E-1\}$ **in parallel do**
6:         Sample skill $\pi_t^e \sim \pi_{\text{meta}}$
7:         With probability $\epsilon$: $a_t^e \sim \text{Unif}$, else $a_t^e \sim \pi_t^e$
8:         Sample rewards $r_t^e \sim P_R(s_t^e, a_t^e)$, skill rewards $r_t^{e,n} \sim P_{R,n}(s_t^e, a_t^e)$ for all $n$
9:         Sample next state $s_{t+1}^e \sim P_S(s_t^e, a_t^e)$
10:       $t \leftarrow t+1$
11:     **end for**
12:     **if** $t \bmod U = 0$ **then**
13:         Compute meta $\lambda$-returns $R_{\lambda,t-1}^e$ to $R_{\lambda,t-U}^e$ for all $e$
14:         Compute skill $\lambda$-returns $R_{\lambda,t-1}^{e,n}$ to $R_{\lambda,t-U}^{e,n}$ for all $e, n$
15:         **for** number of epochs **do**
16:             **for** number of minibatches **do**
17:             Sample minibatch $B$ of size $b \leq EU$ from $\{(t-U, 0), \ldots, (t-1, E-1)\}$
18:             Update meta:

$$\phi_{\text{meta}} \leftarrow \phi_{\text{meta}} + \frac{\alpha_t}{2b} \nabla_{\phi_{\text{meta}}} \sum_{(j,\tau) \in B} \left( R_{\lambda,\tau}^j - Q_{\phi_{\text{meta}}}(s_\tau^j) \right)^2$$

19:             Update skills:

$$\phi_n \leftarrow \phi_n + \frac{\alpha_t}{2b} \nabla_{\phi_n} \sum_{(j,\tau) \in B} \left( R_{\lambda,\tau}^{j,n} - Q_{\phi_n}(s_\tau^{j,n}) \right)^2, \quad \forall n \in \{1, \ldots, N\}$$

20:         **end for**
21:         **end for**
22:     **end if**
23: **end for**

---

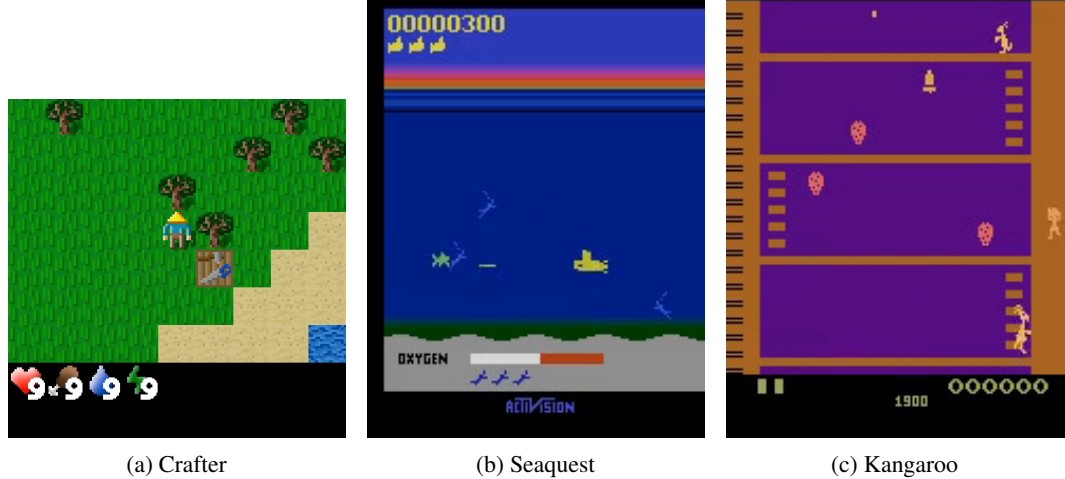

(a) Crafter          (b) Seaquest          (c) Kangaroo

Figure 10: The evaluation environments: Crafter, Seaquest and Kangaroo.

We provide an anonymized code repository[2] that includes all necessary code and config files to reproduce the results from the experiments section, including plots.

### D.1 ENVIRONMENTS

Screenshots of the environments Crafter, Seaquest and Kangaroo are available in Figure 10.

### D.2 EVALUATION METRICS

We empirically evaluate agent performance using three metrics: (1) **Skill Returns** to test whether the skills were learned successfully, (2) **Human-Normalized Score** (HNS) for absolute performance relative to human and random baselines and (3) **Aligned Environment Goal Scores** that measure performance based on the main goals described in the game's manual.

**Human Normalized Score.** Human-Normalized Score standardizes agent performance across Atari environments by accounting for differences in reward scales (Mnih et al., 2015; Machado et al., 2018). Given the average agent score $A$, human score $H$, and random score $R$, HNS is defined as:

$$\text{HNS} = \frac{A - R}{|H - R|}$$

A value of 1.0 indicates human-level performance, values greater than 1.0 indicate superhuman performance, and values below 0 denote sub-random behavior. We adopt the human and random baselines from Badia et al. (2020), derived from professional human play.

**Aligned Environment Goal Scores.** Environment reward signals may not always align with the intended task objectives and can be susceptible to reward hacking. In such cases, agents may learn high-reward behaviors that are non-intuitive and deviate from human-like solutions. To better capture progress toward the actual environment goals, we define two aligned goal-based metrics grounded in the objectives stated in the game manuals.

For *Seaquest*, the goal is to rescue as many divers as possible; for *Kangaroo*, it is to help the mother kangaroo reach and rescue her baby, located on the topmost platform. Accordingly, we track the number of divers retrieved and the number of platforms reached, respectively.

## E ADDITIONAL RESULTS

To assess the generality of NEXUS, we extend our evaluation to three additional Atari environments: *Pong*, *Breakout*, and *Freeway*. Skill learning curves for these environments are shown in Figure 11,

---

[2]https://anonymous.4open.science/r/symbolic_options-302C/

Table 1: Atari Hyperparameters

| Parameter | PQN(-based) | PPO |
|---|---|---|
| Total Timesteps | $5 \times 10^7$ | $5 \times 10^7$ |
| Num Environments | 1024 | 128 |
| Num Steps per Update | 128 | 128 |
| Learning Rate | $1.0 \times 10^{-4}$ | $2.5 \times 10^{-4}$ |
| Max Grad Norm | 10 | 0.5 |
| Discount Factor ($\gamma$) | 0.99 | 0.99 |
| GAE Lambda ($\lambda$) | 0.65 (0.5 with learned meta) | 0.95 |
| GAE Meta Lambda ($\lambda$) | 0.9 | – |
| Num Epochs | 5 | 2 |
| Num Minibatches | 128 | 4 |
| Hidden Size | 64 | – |
| Num Layers | 3 | – |
| Normalization | Layer Norm | – |
| Clip $\epsilon$ | – | 0.2 |
| Entropy Coef | – | 0.01 |
| Value Function Coef | – | 0.5 |
| Anneal LR | False | True |
| $\epsilon$-Start/End/Decay | 1.0 / 0.1 / 0.3 | – |
| Meta $\epsilon$-Start/End/Decay | 1.0 / 0.001 / 0.3 | – |

Table 2: Crafter Hyperparameters

| Parameter | PQN(-based) |
|---|---|
| Total Timesteps | $1 \times 10^9$ |
| Num Environments | 512 |
| Num Steps per Update | 128 |
| Learning Rate | $3.0 \times 10^{-4}$ |
| Max Grad Norm | 0.5 |
| Discount Factor ($\gamma$) | 0.99 |
| GAE Lambda ($\lambda$) | 0.5 |
| GAE Meta Lambda ($\lambda$) | 0.5 |
| Num Epochs | 4 |
| Num Minibatches | 4 |
| Hidden Size | 512 |
| Num Layers | 1 |
| Normalization | Layer Norm |
| Anneal LR | True |
| $\epsilon$-Start/End/Decay | 1.0 / 0.005 / 0.1 |
| Meta $\epsilon$-Start/End/Decay | 1.0 / 0.005 / 0.1 |

while comparisons to baseline agents and ablations—evaluated via human-normalized scores—are presented in Figure 12. Note that these games are generally less complex than *Kangaroo* and *Seaquest*, and the learned skills are not strictly necessary to achieve the environment goals. In particular, the skills "Move Up" and "Avoid Crash" in *Freeway* largely correspond to atomic actions such as `forward` or `noop`. As such, a simple fixed meta-policy operating directly on primitive actions could suffice for solving this task.

**Neuro-symbolic RL baselines.** For improved comparison, we also provide baseline scores of existing interpretable and neuro-symbolic methods on both the default games (cf. Figure 13a) and the simplifications (cf. Figure 13b). The methods are NUDGE (Delfosse et al., 2023a) and BlendRL (Shindo et al., 2025). We also provide a tabular overview of our results and also add the reported scores from SCoBots (Delfosse et al., 2024b) (game simplification scores from Delfosse et al. (2025)) in Table 3.

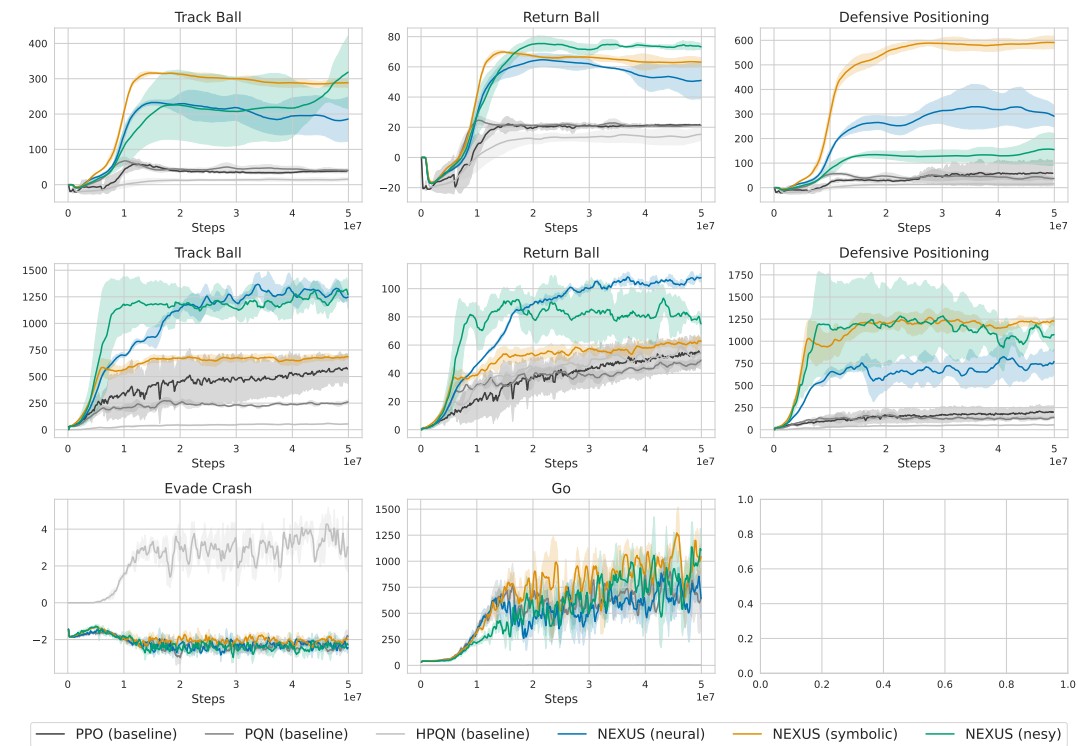

Figure 11: NEXUS successfully learns skills in *Pong* (top), *Breakout* (middle), and *Freeway* (bottom).

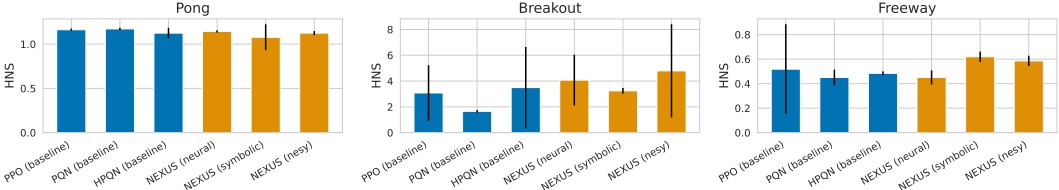

Figure 12: NEXUS achieves performance comparable to baseline methods across the three evaluated environments.

## F  NOISY DETECTIONS

We evaluate the robustness of NEXUS on noise in the object detections both during training and testing. We conduct experiments with 5% and 10% misdetection rate. To emulate the effect of a kalman filter estimating the objects movements during each step, instead of zeroing out the detected objects, we keep the previous time step detections. On top of that, we add gaussian noise with a standard deviation of 3 at each attribute. We show the results with the noisy detection during training in Figure 14. Experiencing noise only during testing is evaluated in Figure 15.

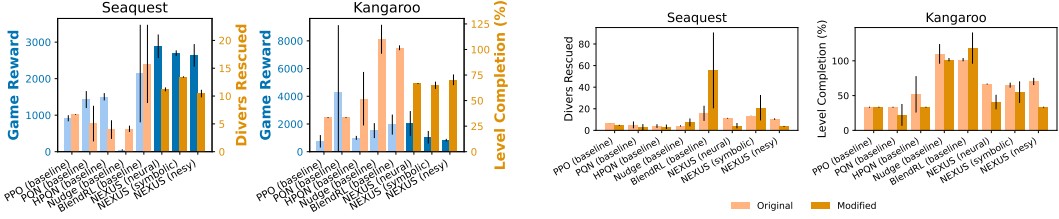

(a) Comparison to other interpretable RL methods on default environments.

(b) Comparison to other interpretable RL methods on simplified games.

Figure 13: Additional neuro-symbolic RL baselines.

Table 3: Performance Comparison on Seaquest and Kangaroo

| Algorithm | Seaquest | | | | Kangaroo | | | |
|---|---|---|---|---|---|---|---|---|
| | Default | Divers Collected | Simplification | Divers Collected | Default | Level Completion (%) | Simplification | Level Completion (%) |
| PPO | $915.3 \pm 82.5$ | $(6.7 \pm 0.1)$ | $0.0 \pm 0.0$ | $(4.7 \pm 0.5)$ | $750.0 \pm 450.0$ | $(33.3 \pm 0.0)$ | $0.0 \pm 0.0$ | $(33.3 \pm 0.0)$ |
| PQN | $1428.3 \pm 232.1$ | $(5.1 \pm 3.2)$ | $0.0 \pm 0.0$ | $(3.0 \pm 2.8)$ | $\mathbf{4300.0 \pm 4833.2}$ | $(33.3 \pm 0.0)$ | $200.0 \pm 141.4$ | $(22.2 \pm 15.7)$ |
| NEXUS (neural) | $\mathbf{2887.6 \pm 322.4}$ | $(11.2 \pm 0.3)$ | $0.0 \pm 0.0$ | $(4.3 \pm 2.5)$ | $2033.3 \pm 899.4$ | $(66.7 \pm 0.0)$ | $666.7 \pm 94.3$ | $(40.7 \pm 10.5)$ |
| NEXUS (symbolic) | $2697.1 \pm 77.7$ | $(13.4 \pm 0.1)$ | $1853.3 \pm 2621.0$ | $(21.0 \pm 11.8)$ | $1042.9 \pm 454.9$ | $(65.0 \pm 3.6)$ | $1346.5 \pm 1042.9$ | $(55.0 \pm 15.6)$ |
| NEXUS (nesy) | $2637.1 \pm 309.9$ | $(10.4 \pm 0.7)$ | $0.0 \pm 0.0$ | $(3.7 \pm 0.5)$ | $833.3 \pm 94.3$ | $(70.4 \pm 5.2)$ | $400.0 \pm 141.4$ | $(33.3 \pm 0.0)$ |
| NUDGE | $46.7 \pm 18.9$ | $(4.1 \pm 0.6)$ | $113.3 \pm 81.3$ | $(7.3 \pm 3.7)$ | $1522.2 \pm 540.5$ | $\mathbf{(110.0 \pm 14.1)}$ | $\mathbf{1966.7 \pm 237.3}$ | $(101.7 \pm 2.4)$ |
| BlendRL | $2138.9 \pm 1335.8$ | $\mathbf{(15.8 \pm 7.0)}$ | $\mathbf{24755.6 \pm 24036.4}$ | $\mathbf{(55.6 \pm 35.1)}$ | $1955.6 \pm 724.9$ | $(101.7 \pm 2.4)$ | $1944.4 \pm 245.5$ | $\mathbf{(118.3 \pm 22.5)}$ |
| SCoBots | $1055.3 \pm 272.6$ | $(- \pm -)$ | $0.0 \pm 0.0$ | $(- \pm -)$ | $2776.6 \pm 1332.4$ | $(- \pm -)$ | $0.0 \pm 0.0$ | $(- \pm -)$ |

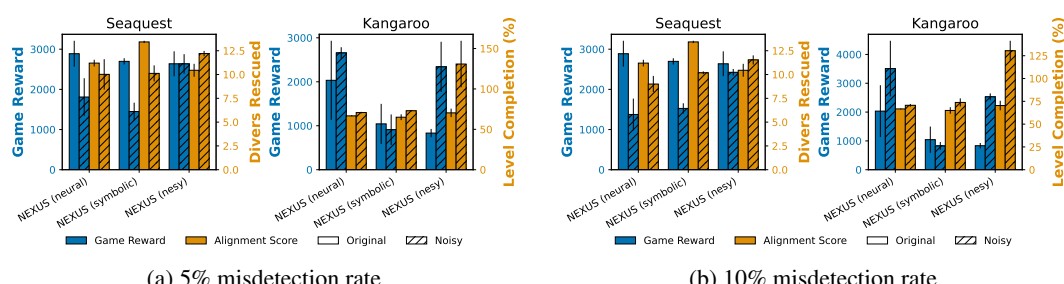

(a) 5% misdetection rate

(b) 10% misdetection rate

Figure 14: Misdetection and noise applied during both training and evaluation.

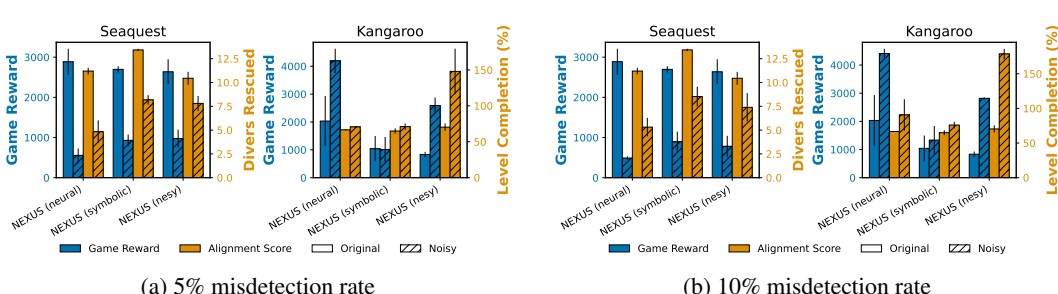

(a) 5% misdetection rate

(b) 10% misdetection rate

Figure 15: Misdetection and noise applied only during evaluation.

## G  LLM INTERACTION

We outline the procedure for leveraging an LLM to generate task-relevant skills, associated reward functions, and a fixed meta-policy rule set. The LLM is conditioned on the game's original manual and structured type information describing the object-centric observations available to the agent.

Initially, the LLM is queried for a set of skills and corresponding reward functions. These outputs can be manually refined before querying the LLM for a meta-policy function that selects which skill to execute. Prior prompts and responses are retained to maintain conversational context, consistent with standard chat behavior. We provide the entire prompt for the game Kangaroo in Figure 16. The prompt for Seaquest was generated equivalently.

Unedited responses from GPT-4o (via chatgpt.com on 21-07-2025) for the games Kangaroo and Seaquest are included in Figure 17–Figure 20.

For Crafter, we used the LLM to generate the list of important tasks, as well as the symbolic meta-policy function. Prompt and answers are available in Figure 21 and Figure 22. Rewards for the specific skills were crafted manually.

Final implementations were modified to align with our framework constraints (e.g., JAX compatibility, indexing conventions). For full details, please refer to the code repository[3].

---

[3]https://anonymous.4open.science/r/symbolic_options-302C/

You are a RL expert and develop a hierarchical agent to play the atari game Seaquest. Below, I have provided a detailed description of the game. Your task is to come up with a short list of essential skills (3-4) that are needed to solve the game.
Game Description:
—
<Atari game manual[a]>
—
Think about what the main goal of this game is, then come up with a list of the required skills to solve the game and provide it in the output. Finally, create a pseudo reward function for each of these skills that a RL agent can use to learn the skills.
The reward functions are called at each step in the environment and retrieve the current and previous symbolic observation of the game. Here is the type definition for the observation:
—

```
class KangarooObservation(NamedTuple):
    player_x: chex.Array
    player_y: chex.Array
    player_o: chex.Array
     platform_positions : chex.Array
     ladder_positions : chex.Array
      fruit_positions : chex.Array
     bell_position : chex.Array
      child_position : chex.Array
      falling_coco_position : chex.Array
    monkey_positions: chex.Array
     coco_positions : chex.Array
```

—
Each skill and it's reward function should be as simple as possible that achieves the wanted behavior.
Provide: List of skills and description, pseudo-code reward functions for each skill.

---
[a]copied entirely from https://www.atariage.com/manual_html_page.php?SoftwareLabelID=923

You are now creating a function that decides at each time step in the game, which skill to choose. The skills are[a]:

- Obstacle Avoidance
- Vertical Navigation
- Fruit Collection
- Goal Reaching (Child Rescue)

The function again has access to the current and previous observation.
Think about at what point in the game you want to use which skill, then provide the function.

---
[a]copied from answer to previous prompt, see Figure 18.

Figure 16: Subsequent prompts (chat-function) to generate a skill-list and corresponding reward functions (top) and meta-policy function (bottom) for the game Kangaroo.

---

### Essential Skills for Solving *Kangaroo* (Atari 2600)

A hierarchical RL agent for *Kangaroo* can be built upon the following four primitive skills:

**1. Obstacle Avoidance**
*Description:* Avoid contact with apples or monkeys via dodging or punching.
*Reward Function:*

```python
def reward_obstacle_avoidance(obs, prev_obs):
    collided = any(np.array_equal(obs.player_x, x) and
                   np.array_equal(obs.player_y, y)
                   for (x, y) in obs.apple_positions + obs.monkey_positions)
    return -1.0 if collided else 0.0
```

**2. Vertical Navigation**
*Description:* Climb ladders to ascend levels and progress.
*Reward Function:*

```python
def reward_vertical_navigation(obs, prev_obs):
    dy = prev_obs.player_y - obs.player_y
    return 0.1 if dy > 0 else 0.0
```

**3. Fruit Collection**
*Description:* Collect fruits and optionally ring a bell to regenerate them.
*Reward Function:*

```python
def reward_fruit_collection(obs, prev_obs):
    reward = 0.0
    if len(obs.fruit_positions) < len(prev_obs.fruit_positions):
        reward += 0.5
    if np.array_equal(obs.player_x, obs.bell_position[0]) and \
       np.array_equal(obs.player_y, obs.bell_position[1]):
        reward += 0.2
    return reward
```

**4. Goal Reaching (Child Rescue)**
*Description:* Reach the child's location to complete the level.
*Reward Function:*

```python
def reward_reach_child(obs, prev_obs):
    return 1.0 if np.array_equal(obs.player_x, obs.child_position[0]) and \
                  np.array_equal(obs.player_y, obs.child_position[1]) else 0.0
```

These modular rewards enable a high-level controller to invoke appropriate behaviors depending on context.

Figure 17: Generated reward functions for Kangaroo by GPT4-O

---

**Kangaroo Skill Selection Function**

To select appropriate skills, the agent follows a fixed priority:

1. **Obstacle Avoidance**: If any apple or monkey is in close proximity.
2. **Goal Reaching**: If the player is near the child on the top platform.
3. **Fruit Collection**: If fruit is visible and reachable.
4. **Vertical Navigation**: Default behavior to explore or ascend.

**Skill Selector Function:**

```python
def select_skill (obs, prev_obs):
    player_pos = np.array ([obs.player_x, obs.player_y])

    def in_danger ():
        danger_entities = obs.apple_positions + obs.monkey_positions
        return any(np.linalg.norm(player_pos - np.array([x, y])) <= 1.0
                   for (x, y) in danger_entities )

    def near_child ():
        child_pos = np.array([obs.child_position[0], obs.child_position[1]])
        return np.linalg.norm(player_pos - child_pos) <= 1.5

    def fruit_visible ():
        return any(abs(y - obs.player_y) <= 1 and
                   abs(x - obs.player_x) <= 3
                   for (x, y) in obs.fruit_positions )

    if in_danger ():
        return obstacle_avoidance
    elif near_child ():
        return goal_reaching
    elif fruit_visible ():
        return fruit_collection
    else :
        return vertical_navigation
```

This rule-based selection policy balances safety, progression, and reward acquisition, enabling effective hierarchical control.

Figure 18: Generated fixed meta-policy function for Kangaroo by GPT4-O

---

## Essential Skills for Solving *Seaquest* (Atari 2600)

To solve *Seaquest* effectively, a hierarchical RL agent must master the following four core skills:

**1. Avoid Obstacles (Survival Navigation)**
*Description:* Avoid collisions with sharks, enemy submarines, missiles, and the patrol sub, both underwater and when surfacing.
*Reward Function (pseudo-code):*

```python
def avoid_obstacles_reward(prev_obs, curr_obs):
    if player_collided(curr_obs):
        return -1.0
    return 0.0

def player_collided(obs):
    return check_collision_with_sharks(obs) or \
           check_collision_with_subs(obs) or \
           check_collision_with_missiles(obs) or \
           check_collision_with_surface_sub(obs)
```

**2. Collect Divers**
*Description:* Navigate to and collect divers to accumulate score and progress.
*Reward Function (pseudo-code):*

```python
def collect_divers_reward(prev_obs, curr_obs):
    prev_count = count_nearby_divers(prev_obs)
    curr_count = count_nearby_divers(curr_obs)
    return float(prev_count > curr_count) * 1.0

def count_nearby_divers(obs):
    return sum(is_near(obs.player_x, obs.player_y, dx, dy)
               for dx, dy in obs.diver_positions)
```

**3. Surface Strategically**
*Description:* Surface safely when carrying six divers or when oxygen is low, to drop off divers and replenish oxygen.
*Reward Function (pseudo-code):*

```python
def surface_reward(prev_obs, curr_obs):
    surfaced_now = prev_obs.player_y > 0 and curr_obs.player_y == 0
    has_6_divers = num_divers_collected(prev_obs) == 6
    low_oxygen = prev_obs.oxygen < OXYGEN_THRESHOLD
    if surfaced_now and (has_6_divers or low_oxygen):
        return 2.0 if has_6_divers else 0.5
    return 0.0

def num_divers_collected(obs):
    return 6 - sum(is_valid(dx, dy) for dx, dy in obs.diver_positions)
```

**4. Eliminate Threats (Combat)**
*Description:* Fire missiles to destroy sharks and enemy submarines, clearing paths and increasing score.
*Reward Function (pseudo-code):*

```python
def eliminate_threats_reward(prev_obs, curr_obs):
    num_enemies_prev = count_enemies(prev_obs)
    num_enemies_curr = count_enemies(curr_obs)
    return float(num_enemies_curr < num_enemies_prev) * 0.5

def count_enemies(obs):
    return sum(is_valid(dx, dy) for dx, dy in obs.shark_positions) + \
           sum(is_valid(dx, dy) for dx, dy in obs.sub_positions)
```

---

### Seaquest Skill Selection Function

Here is a skill selection function that chooses the appropriate control skill at each time step based on game context.

**Skill Selection Function (pseudo-code):**

```python
def select_skill (prev_obs, curr_obs):
    # Skill names
    AVOID = "obstacle_avoidance"
    COLLECT = "collect_divers"
    SURFACE = "surface_strategically"
    COMBAT = "eliminate_threats"

    # === Utility Functions ===
    def is_threat_near (obs: SeaquestObservation) -> bool:
        return check_collision_with_sharks (obs) or \
               check_collision_with_subs (obs) or \
                check_collision_with_missiles (obs) or \
                check_collision_with_surface_sub (obs)

    def num_divers_collected (obs: SeaquestObservation) -> int:
        return 6 - sum(is_valid (x, y) for x, y in obs.diver_positions)

    def low_oxygen(obs: SeaquestObservation) -> bool:
        return obs.oxygen < 20  # adjustable threshold

    def at_surface (obs: SeaquestObservation) -> bool:
        return obs.player_y == 0

    # === Skill Selection Logic ===
    if is_threat_near (curr_obs):
        return AVOID

    if (num_divers_collected (curr_obs) == 6 or low_oxygen(curr_obs)) and \
        not at_surface (curr_obs):
        return SURFACE

    if has_nearby_enemies(curr_obs):  # e.g. visible on screen in front
        return COMBAT

    return COLLECT
```

**Explanation of Logic:**

- **Obstacle Avoidance** is triggered immediately when any nearby threat is detected.
- **Surface Strategically** is invoked if the agent has collected 6 divers or is low on oxygen, and is not already at the surface.
- **Eliminate Threats** is used when enemies are visible but not an immediate threat.
- **Collect Divers** is the default skill when no critical conditions are active.

This rule-based selection mechanism allows a high-level controller to choose among primitive skills in a safety-first manner while still enabling reward-driven exploration and progress.

Figure 20: Generated fixed meta-policy for Seaquest by GPT4-O

You are a reinforcement learning specialist. The following is the basic description of the game Crafter, which you are trying to solve.
—-
Crafter is a game about exploring dungeons, mining, crafting and fighting enemies. The player can move in the four cardinal directions using WASD and can interact using SPACE. Interacting can cause the player to attempt to mine (a block), attack (a creature), drink (water or from a fountain), or eat (fruit).
The player has the 'intrinsics' health, hunger, thirst and energy. Hunger, thirst and energy will naturally decrease and must be replenished by eating, drinking and sleeping respectively. Health will recover when hunger, thirst and energy are non-zero and will decrease if any of these are 0. If the players health falls beneath 0 they will die and the game will restart.
—-
First, come up with 5 basic skills that can be used to solve the game. Only provide necessary skills without duplications. After that, please come up with a simple function (pseudo code) that gets the game-state as input and returns an integer deciding on the active skill to select in the current state. The skills correspond to the ones that you chose before. Try to solve the game as far as possible.
The game-state looks like this:

```python
@struct.dataclass
class Inventory:
    wood: int = 0
    stone: int = 0
    coal: int = 0
    iron: int = 0
    diamond: int = 0
    sapling: int = 0
    wood_pickaxe: int = 0
    stone_pickaxe: int = 0
    iron_pickaxe: int = 0
    wood_sword: int = 0
    stone_sword: int = 0
    iron_sword: int = 0

@struct.dataclass
class Mobs:
    position: jnp.ndarray
    health: int
    mask: bool
    attack_cooldown: int

@struct.dataclass
class EnvState:
    map: jnp.ndarray
    mob_map: jnp.ndarray
    player_position: jnp.ndarray
    player_direction: int
    # Intrinsics
    player_health: int
    player_food: int
    player_drink: int
    player_energy: int
    is_sleeping: bool
    # Second order intrinsics
    player_recover: float
    player_hunger: float
    player_thirst: float
    player_fatigue: float
    inventory: Inventory
    zombies: Mobs
    cows: Mobs
    skeletons: Mobs
    arrows: Mobs
    arrow_directions: jnp.ndarray
    growing_plants_positions: jnp.ndarray
    growing_plants_age: jnp.ndarray
    growing_plants_mask: jnp.ndarray
```

Figure 21: Prompt used to generate a skill-list and meta-policy function for Crafter.

---

### Essential Skills for Solving *Crafter*

**Basic Skills.** We identify the following five skills as sufficient building blocks for solving *Crafter*:

1. **Gather Resources:** Mine wood, stone, coal, iron, or other materials when available.
2. **Craft Tools/Weapons:** Use gathered resources to craft pickaxes and swords for efficiency and combat readiness.
3. **Eat/Drink/Sleep:** Replenish food, water, or energy to maintain survival.
4. **Fight/Defend:** Attack hostile mobs when they are nearby and health is sufficient.
5. **Explore/Move:** Traverse unexplored areas to discover resources, water, and safe locations.

**Policy Function.** The following pseudo-code outlines a rule-based strategy for selecting the active skill based on the current game state.

```python
def select_skill(state: EnvState) -> int:
    """
    Returns the skill index:
    0 = Gather Resources
    1 = Craft Tools/Weapons
    2 = Eat/Drink/Sleep
    3 = Fight/Defend
    4 = Explore/Move
    """

    # --- Survival priority ---
    if state.player_food <= 2 or state.player_drink <= 2 or state.
        player_energy <= 2:
        return 2 # Eat/Drink/Sleep

    # --- Combat ---
    if mob_nearby(state) and state.player_health > 3:
        return 3 # Fight/Defend

    # --- Resource management ---
    if resources_nearby(state) and inventory_needs_tools(state.
        inventory):
        return 0 # Gather Resources

    if can_craft_better_tools(state.inventory):
        return 1 # Craft Tools/Weapons

    # --- Default action ---
    return 4
```

Figure 22: Generated list of essential skills and a rule-based policy for solving *Crafter* by GPT4-O. The policy prioritizes survival, then combat, followed by resource gathering and crafting, and defaults to exploration otherwise.

