# OpenReview forum: "From Objects to Skills: Interpretable Meta-Policies for Neural Control"
_ICLR.cc/2026/Conference — Submitted to ICLR 2026_

### Official Review · Reviewer_2Hts · 2025-10-22

**Soundness:** 2
**Presentation:** 2
**Contribution:** 2
**Rating:** 4
**Confidence:** 3

**Summary:**

This paper introduces LENS, a hierarchical RL framework that achieves strong compositional generalization and interpretability. It decouples high-level reasoning from low-level control by using an object-centric perception module, a pre-trained library of neural skills, and a meta-policy that selects which skill to apply to which objects (as tool/target). The key contribution is enabling zero-shot generalization to structurally novel tasks by composing known skills in new ways.

**Strengths:**

*   By design, the agent's high-level decisions are explicit and human-readable, representing a significant advantage over opaque end-to-end policies and facilitating debugging.
*  The experiments convincingly demonstrate the ability to solve complex, multi-step manipulation tasks that were not seen during training, highlighting the framework's practical effectiveness.

**Weaknesses:**

*   The system's success is critically dependent on a high-quality, pre-trained skill library and a flawless perception module. The paper does not fully address the scalability of creating this skill library or how the system handles perception failures.
*   The meta-policy is trained via supervised learning, which requires expert demonstrations for high-level actions. This raises concerns about scalability and autonomy compared to a pure RL approach for discovering high-level strategies.

**Questions:**

How does the system handle failures in underlying modules? For instance, what is the recovery mechanism if a low-level skill fails to execute or the perception module makes an error?

**Details Of Ethics Concerns:**

No concern.

---

> ### Author Response · Authors · 2025-11-21
>
> We thank the reviewer for their time and appreciate the recognition of the significant advantage of NEXUS over neural policies and are happy to address their concerns.
>
> **All changes incorporated are highlighted blue in the paper.**
>
> 1. **Dependence on pre-trained skill library and flawless perception module.**
>     - One of the core contributions of NEXUS is that it does not depend on a pre-trained skill library but rather learns them during training with generated reward functions. We provide a detailled evaluation of this in 4.2 (Disentangled skill learning, Q1) and Figures 3 and 4.
>
> 2. **Flawless perception module and handling of perception failures.**
>
>     - While we agree that misdetection is a pressing issue, it does not invalidate this work. We use a perfect object extraction method to isolate the quality of the object-centric policy, as many other object-centric RL work do [7, 8, 9]. Thus, any limitation can be attributed to NEXUS rather than to an object extraction module. Of course, perception remains a highly relevant field, and substantial progress has been demonstrated. For example, SAM2 [1] (semi-supervised) and DINOv3 [2] (self-supervised) become increasingly good at segmenting real world video data (up to 90% SAM2 / 71% DINOv3 J&F [region similarity and contour-based accuracy]). For Atari games, there exists various works focusing on extracting objects (and their attributes) from the games, which achieve impressive results [3, 4, 5, 6, 7]. We also added these points to the limitation section of the paper (page 9, line 461+).
>    - However, as correctly pointed out, object detection is still not entirely solved, so we decided to run an **additional experiment** to evaluate NEXUS under noisy detection. This includes four evaluations, with either just evaluation or training and evaluation with a 5% and 10% misdetection rate and a std-deviation of 3 pixels on each detection. If a misdetection happens, we use the previous frame's detection as fill-in, unless 4 misdetection happen in a row, which will lead to the object being set to zero entirely. The method could probably be further improved by tracking the objects, e.g. using a kalman filter.
>     - Interestingly, while the additional noise leads to a performance drop for the neural and the symbolic meta-policies in Seaquest (as expected), the neuro-symbolic one is more robust. Additionally, our results on Kangaroo indicate that adding noise can even have a positive impact on the performance.
>
>    - We also incorporated the results in the paper (page 8, line 415+), with full details (and all experiments) in the Appendix F.
>
> 3. **Suspected supervised learning of the meta-policy.**
>     - None of the three meta-policy variations in NEXUS require supervised learning or expert demonstrations. It is either defined a priori (e.g. by a LLM) in the case of the (neuro-)symbolic variations or learned autonomously in the neural version.
>
> **Questions:**
>
> 1. How does the system handle failures in underlying modules? For instance, what is the recovery mechanism if a low-level skill fails to execute or the perception module makes an error?
>     - We thank the reviewer for raising this critical point regarding system robustness. The meta-policy in NEXUS selects a skill at every timestep to produce the next low-level action. If a selected skill fails, the resulting state at the following timestep reflects this failure. The meta-policy then reassigns control by choosing an alternative (or the same) skill based on learned Q-values, symbolic rules, or their combination.
>     - As explained above, perception errors will impact the performance of the neural and symbolic meta-policy variations, but neuro-symbolic NEXUS is quite robust against detection noise and errors.
>
> —------------------------------------------------------------------------------------------------------------
>
> [1] Ravi, Nikhila, et al. "SAM 2: Segment Anything in Images and Videos." The Thirteenth International Conference on Learning Representations.
>
> [2] Siméoni, Oriane, et al. "Dinov3." arXiv (2025).
>
> [3] Li, Yuezhang, Katia Sycara, and Rahul Iyer. "Object-sensitive deep reinforcement learning." arXiv preprint arXiv:1809.06064 (2018).
>
> [4] Lin, Zhixuan, et al. "SPACE: Unsupervised Object-Oriented Scene Representation via Spatial Attention and Decomposition." International Conference on Learning Representations.
>
> [5] Delfosse, Quentin, et al. "Boosting object representation learning via motion and object continuity." Joint European Conference on Machine Learning and Knowledge Discovery in Databases. Cham: Springer Nature Switzerland, 2023.
>
> [6] Smirnov, Dmitriy, et al. "Marionette: Self-supervised sprite learning." Advances in Neural Information Processing Systems 34 (2021).
>
> [7] Delfosse, Quentin, et al. "Interpretable and explainable logical policies via neurally guided symbolic abstraction." Advances in Neural Information Processing Systems 36 (2023).

---

> > ### Comment · Reviewer_2Hts · 2025-11-22
> >
> > Thank you for your reply. I still have doubts about the source of NEXUS's prior knowledge and its practical applicability. If you can demonstrate the effectiveness of NEXUS on continuous control tasks (e.g., HumanoidBench [1]), I will increase my score.
> >
> > [1] Sferrazza, Carmelo, et al. "Humanoidbench: Simulated humanoid benchmark for whole-body locomotion and manipulation." arXiv preprint arXiv:2403.10506 (2024).

---

> ### Author Response · Authors · 2025-11-24
>
> 1. **Usage of prior knowledge is described in the paper**:
>     - Introduction (top of page 2)
>     - Section 3 (page 3, line 147+)
>     - Section 3.2 (page 4, line 196+)
>     - Section 3.3 (page 5, line 224+)
>     - Section 4.2 (page 6, line 316+, line 350+, line 361+)
>     - Appendix, Section G (page 23+)
>
> For increased clarity, **we added a summary** to the method (page5, line 242) and additional reasoning in the limitations section.
>
> 2. While applying NEXUS to **continuous control tasks is an exciting direction**, we respectfully position this as **future work** for these reasons:
>     - The core contribution of this work is interpretable, hierarchical decision making and effective utilization of prior knowledge. This is **fully validated by our experiments in discrete action spaces**. The complexity of the action space (discrete vs. continuous) is **orthogonal to this work**.
>     - As noted in our limitations (Page 9, lines 477+), extending NEXUS to continuous spaces is **theoretically straightforward**. It is a standard architectural modification (e.g. going from DQN to SAC) that **does not alter the fundamental logic of the NEXUS** framework.
>    - The current suite of experiments is **sufficient to validate the claims** made in the paper.

---

### Official Review · Reviewer_ebVk · 2025-11-01

**Soundness:** 1
**Presentation:** 2
**Contribution:** 2
**Rating:** 2
**Confidence:** 3

**Summary:**

This paper presents NEXUS, a hierarchical structure of neuro-symbolic RL, which uses LLMs to identify skills for a given task and compose reward function for learning neural policies regarding each skill. Specifically, before policy learning NEXUS identifies the pre-conditions for each skill using a LLM, what are used to activate skills for each state. Crucially, both the reward function and the pre-conditions are defined for concepts/symbols at each state, provided by the simulator. The authors evaluate NEXUS on a few atari games and compares it with its neural counterpart, PQN and PPO.

**Strengths:**

1. The paper presents effort in learning disentangled skills, such as "Rescue Divers" and "Shoot Enemy" for the Atari game Seaquest. Such a design sheds light on more effect usage of neural policies for neuro-symbolic RL.

2. The paper showcases that LLM-generated reward functions are quite competitive when visual symbols can be reliably obtained.

**Weaknesses:**

1. While using LLMs to identify pre-conditions and reward functions for learning hierarchical policies, the paper does not include a discussion for prior attempt on LLM-generated rules or rewards for NSRL.

2. No performance comparison with other neuro-symbolic approach is presented, even though Atari is a common benchmark for this line of work.

3. The paper lacks a discussion for the failure patterns of LLM-generated rules or rewards.

4. The entire framework is based on the assumption that visual concepts can be reliably identified at real-time, which is quite impractical, and there is no discussion for how to address such assumption.

**Questions:**

1. Since both the LLM-generated rules and rewards are not updated during policy learning, how do you ensure their alignments with the task of interest and eliminate the influence of hallucination?

2. Is is possible to apply this method on benchmarks other than OCAtari, especially those that do not provide ground-truth visual concepts?

---

> ### Author Response · Authors · 2025-11-21
>
> We thank the reviewer for their work and for recognizing the novelty of our disentangled skill design. We have carefully considered the questions raised and address them below.
>
> **All changes incorporated are highlighted blue in the paper.**
>
> 1. **Missing discussion of prior LLM-generated rules and rewards.**
>    - We thank the author for pointing out the previously limited discussion on LLM-generated rules and rewards and incorporated a paragraph on this topic in the Limitations section (page9, line 466+). Please let us know if we missed any additional literature on this topic.
>
> 2. **Performance comparison to other neuro-symbolic approaches.**
>
>     - We agree that prior to this discussion, we were missing comparison to other interpretable RL methods, and thus provide **additional baseline experiments** with NUDGE [8] and BlendRL [9]. We also added the reported scores from SCoBots [10] (Appendix Section E, pages 19/20/21). If there are any other SOTA interpretable RL methods missing, please let us know so we can add them.
>
> 3. **The paper lacks a discussion for the failure patterns of LLM-generated rules or rewards.**
>
>    - We agree that analyzing LLM-related failure patterns for rules or rewards is a compelling line of research. However, the primary contribution of this work is the introduction and validation of NEXUS, demonstrating that LLMs can effectively assist in creating interpretable neuro-symbolic systems. We defer a deeper analysis of failure modes and robustness to subsequent dedicated studies, which we have already started working on.
>
> 4. **Assumption of reliable visual concept detection.**
>
>     - While we agree that misdetection is a pressing issue, it does not invalidate this work. We use a perfect object extraction method to isolate the quality of the object-centric policy, as many other object-centric RL work do [7, 8, 9]. Thus, any limitation can be attributed to NEXUS rather than to an object extraction module. Of course, perception remains a highly relevant field, and substantial progress has been demonstrated. For example, SAM2 [1] (semi-supervised) and DINOv3 [2] (self-supervised) become increasingly good at segmenting real world video data (up to 90% SAM2 / 71% DINOv3 J&F [region similarity and contour-based accuracy]). For Atari games, there exists various works focusing on extracting objects (and their attributes) from the games, which achieve impressive results [3, 4, 5, 6, 7]. We also added these points to the limitation section of the paper (page 9, line 461+).
>
>    - However, as correctly pointed out, object detection is still not entirely solved, so we decided to run an **additional experiment** to evaluate NEXUS under noisy detection. This includes four evaluations, with either just evaluation or training and evaluation with a 5% and 10% misdetection rate and a std-deviation of 3 pixels on each detection. If a misdetection happens, we use the previous frame's detection as fill-in, unless 4 misdetection happen in a row, which will lead to the object being set to zero entirely. The method could probably be further improved by tracking the objects, e.g. using a kalman filter.
>
>    - Interestingly, while the additional noise leads to a performance drop for the neural and the symbolic meta-policies in Seaquest (as expected), the neuro-symbolic one is more robust. Additionally, our results on Kangaroo indicate that adding noise can even have a positive impact on the performance.
>     - We also incorporated the results in the paper (page 8, line 415+), with full details (and all experiments) in the Appendix F. Thanks again for pointing this out.

---

> > ### Author Response · Authors · 2025-11-21
> >
> > **Questions:**
> >
> > 1. Since both the LLM-generated rules and rewards are not updated during policy learning, how do you ensure their alignments with the task of interest and eliminate the influence of hallucination?
> >
> >     - We address the alignment and hallucination concerns through two key mechanisms: contextual grounding and system structure.
> >     - **Contextual Grounding**: We assume the task goals are known a priori. The LLM's role is not to invent the task, but to translate the high-level task description into formal, symbolic components (rules and rewards). Crucially, we significantly mitigate potential hallucination errors by explicitly providing type information for the observations and the complete list of generated skills in the subsequent prompts. This grounding makes the LLM’s output more specific and factual. We have updated the prompts in the Appendix (Section G, page 23+) to fully reflect these crucial contextual additions.
> >     - **Structural Separation**: Since the LLM provides the design (the interpretable symbolic policy and reward), and the neural components handle the optimization process, the LLM’s output does not need to be updated during policy learning. The symbolic layer remains fixed, ensuring consistency while the neural component learns to satisfy its requirements. Changes to the task would indeed necessitate a new LLM-assisted design phase (i.e., new prompts).
> >     - Furthermore, both are potentially verifiable by an expert, as the meta policy and the reward are symbolic and thus transparent and concise (thanks to the hierarchical structure of NEXUS).
> >
> > 2. Is it possible to apply this method on benchmarks other than OCAtari, especially those that do not provide ground-truth visual concepts?
> >
> >    - The extension of NEXUS to domains lacking ground-truth symbolic concepts (like many standard continuous control benchmarks) is a critical direction. While NEXUS currently relies on an object-centric perception module, this dependency is modular and external to the core neuro-symbolic reasoning.
> >     - In principle, the system can integrate any state-of-the-art method for object-centric representation learning or symbol grounding. These methods, which have seen significant advancements recently, can be used to generate the necessary representations from raw pixel input.
> >     - However, we acknowledge that coupling NEXUS with such a complex learned perception module introduces new challenges that must be tested empirically. As described above, we have chosen to focus this initial work on validating the core neuro-symbolic interaction under ideal conceptual conditions. Follow-up work is already planned to rigorously test the application of NEXUS in conjunction with learned perception modules on more complex, pixel-based benchmarks.
> >
> > -----------------------------------------------------------------------------------------------------
> > [1] Ravi, Nikhila, et al. "SAM 2: Segment Anything in Images and Videos." The Thirteenth International Conference on Learning Representations.
> >
> > [2] Siméoni, Oriane, et al. "Dinov3." arXiv (2025).
> >
> > [3] Li, Yuezhang, Katia Sycara, and Rahul Iyer. "Object-sensitive deep reinforcement learning." arXiv preprint arXiv:1809.06064 (2018).
> >
> > [4] Lin, Zhixuan, et al. "SPACE: Unsupervised Object-Oriented Scene Representation via Spatial Attention and Decomposition." International Conference on Learning Representations.
> >
> > [5] Delfosse, Quentin, et al. "Boosting object representation learning via motion and object continuity." Joint European Conference on Machine Learning and Knowledge Discovery in Databases. Cham: Springer Nature Switzerland, 2023.
> >
> > [6] Smirnov, Dmitriy, et al. "Marionette: Self-supervised sprite learning." Advances in Neural Information Processing Systems 34 (2021).
> >
> > [7] Delfosse, Quentin, et al. "Interpretable and explainable logical policies via neurally guided symbolic abstraction." Advances in Neural Information Processing Systems 36 (2023).
> >
> > [8] Shindo, Hikaru, et al. "BlendRL: A Framework for Merging Symbolic and Neural Policy Learning." The Thirteenth International Conference on Learning Representations.
> >
> > [9] Delfosse, Quentin, et al. "Interpretable concept bottlenecks to align reinforcement learning agents." Advances in Neural Information Processing Systems 37 (2024).
> >
> > [10] Xie, Tianbao, et al. "Text2Reward: Reward Shaping with Language Models for Reinforcement Learning." The Twelfth International Conference on Learning Representations (2023).

---

### Official Review · Reviewer_pQ3D · 2025-11-02

**Soundness:** 3
**Presentation:** 4
**Contribution:** 4
**Rating:** 8
**Confidence:** 4

**Summary:**

This paper tackles the challenge of making RL agents more interpretable and robust by **reasoning at the level of skills rather than low-level actions**. The authors propose NEXUS, a hierarchical framework that combines neural skill controllers with symbolic or neuro-symbolic meta-policies to select which skill to execute based on object-centric state representations. By integrating symbolic reasoning with learned control, NEXUS enables transparent decision-making while maintaining flexibility and performance. Experiments on several Atari-style and Crafter environments show that NEXUS learns distinct, meaningful skills, achieves competitive rewards, avoids reward hacking, and generalizes better to distribution shifts compared to neural baselines.

**Strengths:**

- Experiments are very thoughtfully designed to support the key claims about interpretability, robustness, and performance.
- The perspective of shifting interpretability  from the action level to the skill level is novel and creative.
- The paper is well-written and conceptually clear, with motivating examples, intuitive figures to follow along.

**Weaknesses:**

- Although some skill rewards are LLM-generated, the framework still relies on meaningful skill definitions provided a priori.
- Along the same point, while symbolic meta-policies are interpretable, it’s not evident how their rule sets scale with larger skill repertoires or more complex environments. The evaluation too mainly focuses on 2D domains.
- The paper could engage more directly with established hierarchical RL frameworks to clarify what is genuinely new in the meta-policy formulation beyond symbolic labeling.

**Questions:**

- To what extent can the framework autonomously discover useful skills, rather than relying on manually defined or LLM-suggested ones? Could the meta-policy guide skill discovery dynamically during training? How does the system behave when symbolic rules conflict or are incorrect?
- Since object-centric state representations are assumed, how sensitive is performance to noise or inaccuracies in object detection?
- Have the authors considered any quantitative or user-centered evaluation of interpretability (e.g., human predictability or trust metrics)?
- how might NEXUS extend to continuous control or robotics tasks where symbolic conditions and skills are less clear?

---

> ### Author Response · Authors · 2025-11-21
>
> We are thankful for the thoughtful review and are pleased that the reviewer recognized the significance of moving from atomic actions to skills. We have carefully considered the questions raised and address them below.
>
> **All changes incorporated are highlighted blue in the paper.**
>
> 1. **Although some skill rewards are LLM-generated, the framework still relies on meaningful skill definitions provided a priori.**
>
>     - Actually, NEXUS does not rely on a priori definition of meaningful skill definitions. Conditioned on the game manual, we query the LLM to provide a list of necessary skills together with the reward functions. We acknowledge the previous vagueness in the provided prompts and update them accordingly (Appendix G, page 23+).
>
> 2. **The paper could engage more directly with established hierarchical RL frameworks to clarify what is genuinely new in the meta-policy formulation beyond symbolic labeling.**
>
>     - Thanks for pointing out the unclear novelty compared to HRL. We discuss the difference to the Options framework [1] before concluding. To improve clarity we added a final summarization of the key innovations to the paragraph (page 10, line 503+). Please let us know if you would like to see any additional hierarchical frameworks compared to NEXUS.
>
> **Questions**:
> 1. To what extent can the framework autonomously discover useful skills, rather than relying on manually defined or LLM-suggested ones?
>     - In its current form, NEXUS does not autonomously discover skills, but an iterative design that re-queries an LLM once a change to the environment has been detected, is planned for follow-up work. We added a sentence to the conclusion (page 10, line 519+).
>
> 2. Could the meta-policy guide skill discovery dynamically during training? How does the system behave when symbolic rules conflict or are incorrect?
>     - The meta-policy could also be adapted by re-querying the LLM. The symbolic meta-policy of NEXUS incorporates ordering, i.e. it would default to the first rule that is evaluated as true. The neuro-symbolic variations, on the other hand, does weigh the rules that evaluate to true (using probabilistic logic) at the same time and will thus choose the most appropriate one for maximizing environment reward. Incorrect rules would typically hinder convergence to optimal solutions.
>
> 3. Since object-centric state representations are assumed, how sensitive is performance to noise or inaccuracies in object detection?
>
>     - We decided to run an **additional experiment** to evaluate NEXUS under noisy detection. This includes four evaluations, with either just evaluation or training and evaluation with a 5% and 10% misdetection rate and a std-deviation of 3 pixels on each detection. If a misdetection happens, we use the previous frame's detection as fill-in, unless 4 misdetection happen in a row, which will lead to the object being set to zero entirely. The method could probably be further improved by tracking the objects, e.g. using a kalman filter.
>
>     - Interestingly, while the added noise leads to a performance drop for the neural and the symbolic meta-policies in Seaquest (as expected), the neuro-symbolic one is more robust. Additionally, the results on Kangaroo indicate that adding noise can even have a positive impact on the performance.
>    -  We also incorporate the results in the paper (page 8, line 415+), with full details (and all experiments) in the Appendix F.
> Thanks for pointing this out.
>
> 4. Have the authors considered any quantitative or user-centered evaluation of interpretability (e.g., human predictability or trust metrics)?
>
>     - We agree that user-centered evaluations (such as human predictability or trust metrics) are the ultimate test for interpretability. However, in this work, our primary focus was to establish the technical feasibility of NEXUS. Inspired by the feedback, we have added a sentence to the Limitations to explicitly state that future work should focus on evaluating human-centered attributes like trust and predictability (Conclusion, page10, line 518+).
>
> 5. How might NEXUS extend to continuous control or robotics tasks where symbolic conditions and skills are less clear?
>
>     - Extending to robotics requires addressing the symbol grounding problem. We propose that the 'unclear' symbolic conditions mentioned can be learned rather than hard-coded. By integrating a trainable perception layer (e.g., Slot Attention or Sparse Autoencoders) directly into the pipeline, NEXUS can treat symbols as latent variables. Additionally, to enable continuous control tasks, the current approach of maximizing the learned Q-values would need replacement with a continuous actor (such as in DDPG or SAC).
>
> —-----------------------------—-----------------------------—------------------------
>
> [1] Sutton, Richard S., Doina Precup, and Satinder Singh. "Between MDPs and semi-MDPs: A framework for temporal abstraction in reinforcement learning." Artificial intelligence 112.1-2 (1999): 181-211.

---

### Official Review · Reviewer_noWc · 2025-11-02

**Soundness:** 3
**Presentation:** 3
**Contribution:** 2
**Rating:** 4
**Confidence:** 4

**Summary:**

The paper proposes NEXUS, a hierarchical reinforcement learning (RL) framework that combines neural skills (low-level policies) with interpretable meta-policies (high-level decision-making) to achieve transparency in control tasks. Drawing from dual-process theory, it structures agents with fast neural execution (System 1) and deliberative symbolic reasoning (System 2). Key components include object-centric representations for symbolic states, LLM-generated reward functions for disentangled skills, and three meta-policy variants: fully neural, fully symbolic, and neuro-symbolic (NeSy). The framework extends Parallelised Q-Networks (PQN) to hierarchical settings, training skills off-policy with skill-specific rewards. Experiments on Atari games (Seaquest, Kangaroo) and Crafter evaluate disentanglement, interpretability, performance, and robustness to simplifications, claiming reduced reward hacking and better generalization.

**Strengths:**

The paper addresses a timely issue in RL: interpretability in hierarchical agents, which is crucial for alignment and debugging. The integration of LLMs for generating rewards and rules is practical, reducing manual effort and enabling adaptation. The neuro-symbolic variant offers a balanced trade-off between flexibility and transparency, with clear visualizations aiding understanding. Experiments show promising results, like balanced skill learning and robustness to environment simplifications, highlighting potential advantages over baselines like PPO and PQN.

**Weaknesses:**

Methodologically, the reliance on pre-extracted object-centric states assumes perfect perception, ignoring real-world challenges like noisy or incomplete object detection, which limits applicability. LLM generation of rewards/rules is underexplored—prompts are vague (Section E), and no analysis of LLM errors or sensitivity to model choice (e.g., GPT-4 vs. others). The hierarchical PQN extension is incremental, building on existing works without novel theoretical contributions. Baselines are weak—HPQN is a strawman without skill rewards—and no comparison to state-of-the-art like neuro-symbolic RL. Robustness claims are overstated; simplifications (e.g., removing enemies) favor symbolic policies by design, but no ablation on rule quality or generalization to harder shifts (e.g., added obstacles).

**Questions:**

While interpretability in RL is important, NEXUS largely recombines established ideas—hierarchical RL, object-centric states, and LLM-aided RL—without breakthroughs. The core innovation (NeSy meta-policy) is a simple mask on Q-values, lacking depth compared to prior neuro-symbolic works. Do I miss anything here?

The motivation is not well motivated. There have been approaches that use symbolic reasoning to help low-level policy learning. What advantages does the proposed approach have compared to previous work, intuitively speaking?

Is the sentence "(1) a high-level meta-policy deciding between" incomplete ?

---

> ### Author Response · Authors · 2025-11-21
>
> We thank the reviewer for the time and effort put into the review and for recognizing the timely issue addressed by NEXUS and the promising results of our evaluation. We would like to address the following points of concern.
> **All changes incorporated are highlighted blue in the paper.**
>
> **1. Assumption of perfect perception.** While we agree that misdetection is a pressing issue, it does not invalidate this work. We use a perfect object extraction method to isolate the quality of the object-centric policy, as many other object-centric RL work do [7, 8, 9]. Thus, any limitation can be attributed to NEXUS rather than to an object extraction module. Of course, perception remains a highly relevant field, and substantial progress has been demonstrated. For example, SAM2 [1] (semi-supervised) and DINOv3 [2] (self-supervised) become increasingly good at segmenting real world video data (up to 90% SAM2 / 71% DINOv3 J&F [region similarity and contour-based accuracy]). In the Atari domain, various works focuse on object extraction (and their attributes) from the games, which achieve impressive results [3, 4, 5, 6]. Any of these methods could be applied on top of NEXUS. We also added these points to the limitation section of the paper (page 9, line 461+).
>
> However, as correctly pointed out, object detection is still not considered solved, so **we run additional experiments** to evaluate NEXUS robustness under noisy detection. We consider four settings involving evaluation only or joint training and evaluation with 5% and 10% misdetection rates together with a Gaussian noise (of std. 3 pixels) on the object position. If a misdetection happens, we use the position extracted from previous frame's detection as fill-in, unless several misdetections happen consecutively, which will lead to the object being set to zero entirely. The method could probably be further improved by tracking the objects, e.g. using a Kalman filter.
> Interestingly, while the additional noise leads to a performance drop for the neural and the symbolic meta-policies in Seaquest (as expected), the neuro-symbolic one is more robust.
> Additionally, our results on Kangaroo indicate that adding noise can even have a positive impact on the performance.
> We incorporated these results in the paper (page 8, line 415+), with full details (and all experiments) in the Appendix E.
> Thanks again for pointing this out.
>
> **2. LLM-generated rewards/rules.** We refined the prompts to address the previously noted ambiguity. The updated versions are provided in Appendix G starting on page 22.
>
> Regarding the under-exploration of LLM-generated rules and rewards: We agree that comparing models and failure modes is a promising avenue for research and are, in fact, actively pursuing it in follow-up work. However, we believe that such a study is out of the scope of this paper, as studying how different LLMs can be leveraged to generate rules or reward function constitute dedicated standalone research works [10, 11, 12]. That said, we acknowledge (as did another reviewer ebVk) that our prior treatment of this topic was too brief. We have updated the manuscript to include a dedicated discussion on the potential and limitations of LLM-assisted rule generation (Page 9, Line 466+).
>
> **3. Baseline methods.** First, we would like to object to the claim that HPQN is a strawman. Our method builds on it, so comparing to it allows us to isolate the contribution of the hierarchical structure alone, which provides a controlled baseline for ablation. This comparison demonstrates that hierarchical decomposition by itself yields limited gains, and that the proposed generation of skill rewards is a critical factor for improving performance. Including this variant is therefore necessary to establish the limitation of hierarchical decomposition without skill rewards.
>
> We appreciate the feedback regarding the missing comparisons to other interpretable RL methods. To address this, we have included **additional baseline experiments** against established methods: NUDGE [7] and BlendRL [8]. Furthermore, we have incorporated the reported scores from SCoBots [9]. We believe these additions significantly strengthen our empirical analysis. If the reviewer can suggest any other specific state-of-the-art interpretable RL methods that would provide a critical comparison, we would be happy to arrange the necessary experiments.
>
> **4. Robustness Overclaim.** We intended to convey that NEXUS exhibits robustness against game simplifications, a capability often lacking in purely neural methods. We agree with the reviewer that this specific claim was not adequately introduced in the abstract, potentially leading to the perception of an overclaim. We have updated the abstract to clearly reflect this core contribution and align it with the detailed discussion in the main body of the paper. Thank you for this essential pointer.

---

> ### Author Response · Authors · 2025-11-21
>
> **5. Novelty / Core innovation / Motivation.** The most important aspect considered in this work is, as outlined in the motivation, the high-level interpretability. Existing interpretable RL methods are only transparent, but explode in size, which makes it difficult for humans to follow the reasoning process. To improve this point in our paper, we included additional examples in the appendix (Section B).
>
> With NEXUS, we are able to show that hierarchical RL, with disentangled skills (by LLM-generated rewards), resolves this issue conveniently. Crucially, hierarchical RL without the skill-specific reward functions is not sufficient, as the resulting policies are entangled (thus not interpretable) and less performant (HPQN baseline).
>
> We acknowledge that the individual components (HRL, LLMs for reward generation) are known. However, their integration into a simple, unified pipeline to yield improved high-level interpretability and robustness to environment simplifications is, to our knowledge, novel. Methodological simplicity does not preclude novelty.
>
> **Sentence structure question:** Thanks for pointing this out. While the sentence was not incomplete, it did read a bit odd, so we changed it accordingly.
>
> —----------------------------—----------------------------—----------------------------—----------------------------—------------------------
>
> [1] Ravi, Nikhila, et al. "SAM 2: Segment Anything in Images and Videos." The Thirteenth International Conference on Learning Representations.
>
> [2] Siméoni, Oriane, et al. "Dinov3." arXiv (2025).
>
> [3] Li, Yuezhang, Katia Sycara, and Rahul Iyer. "Object-sensitive deep reinforcement learning." arXiv preprint arXiv:1809.06064 (2018).
>
> [4] Lin, Zhixuan, et al. "SPACE: Unsupervised Object-Oriented Scene Representation via Spatial Attention and Decomposition." International Conference on Learning Representations.
>
> [5] Delfosse, Quentin, et al. "Boosting object representation learning via motion and object continuity." Joint European Conference on Machine Learning and Knowledge Discovery in Databases. Cham: Springer Nature Switzerland, 2023.
>
> [6] Smirnov, Dmitriy, et al. "Marionette: Self-supervised sprite learning." Advances in Neural Information Processing Systems 34 (2021).
>
> [7] Delfosse, Quentin, et al. "Interpretable and explainable logical policies via neurally guided symbolic abstraction." Advances in Neural Information Processing Systems 36 (2023).
>
> [8] Shindo, Hikaru, et al. "BlendRL: A Framework for Merging Symbolic and Neural Policy Learning." The Thirteenth International Conference on Learning Representations.
>
> [9] Delfosse, Quentin, et al. "Interpretable concept bottlenecks to align reinforcement learning agents." Advances in Neural Information Processing Systems 37 (2024).
>
> [10] Xie, Tianbao, et al. "Text2Reward: Reward Shaping with Language Models for Reinforcement Learning." The Twelfth International Conference on Learning Representations (2023).
>
> [11] Sun, Shengjie, et al. "A large language model-driven reward design framework via dynamic feedback for reinforcement learning." arXiv (2024).
>
> [12] Xie, Tianbao, et al. "Text2reward: Automated dense reward function generation for reinforcement learning." International Conference on Learning Representations (ICLR) 2024.

---

### Author Response · Authors · 2025-12-02
**TLDR/Summary of the discussion phase**

We thank the reviewers for their constructive feedback. As the discussion phase closes, we provide a brief summary of the major revisions and experimental additions that address the key concerns raised by the reviewers. **We adapted the paper significantly and highlighted the changes in blue.**

## Robustness to Perception Noise

**(addresses reviewers noWc, pQ3D, ebVk, 2Hts).** The main critique was the assumption of "perfect perception" (a choice made to scientifically isolate NEXUS from perception errors) . We added **comprehensive experiments** (Q5) with 5-10% misdetection rates and positional noise (std-dev: 3px).
- **Outcome**: While the purely neural and purely symbolic variants drop in performance, **neuro-symbolic NEXUS remains robust**.

## Additional Neuro-Symbolic Baselines

**(Addresses reviewers noWc, ebVk)** We added comparisons to SOTA interpretable methods (Appendix E): **NUDGE, BlendRL, and SCoBots**. NEXUS remains competitive and simultaneously improves high-level interpretability (see also Motivation and Appendix B).

## LLM Generated Rewards and Discussion:

**(Adresses reviewers noWc, ebVk, 2Hts)** We addressed ambiguity by **refining the LLM prompts** (Appendix G) to show the specific type-constrained grounding used to limit hallucination errors and added a discussion about the **prior knowledge required** (end of chapter 3). We additionally provide a new discussion on the **potential and limitations of LLM-assisted rule generation** in Limitations.

## Reviewer 2Hts (Score: 4)
We respectfully request the AC weigh this review lightly due to **fundamental factual errors** and **goalpost shifting**:
- **Factually Incorrect**: The reviewer claimed NEXUS relies on a "pre-trained skill library" and "supervised meta-policy learning." Both claims are false. As stated in the paper, skills are learned during training and the meta-policy is not supervised.
- **Out of Scope**: In a late comment, the reviewer requested results on HumanoidBench (continuous control). This shifts the goalposts; our paper focuses on hierarchical interpretability in discrete domains (Atari/Crafter), a standard and valid scope for this track.

## Conclusion
Reviewer pQ3D (Score: 8) strongly champions the work for its novel skill-level interpretability. With the perception/noise and baseline concerns empirically resolved and the LLM generation process clarified, we believe the paper is ready for acceptance.

---

### Meta-Review · Area_Chair_zFM6 · 2026-01-12

**Summary:**

The reviewers raised many important concerns and unfortunately the authors and reviewers were unable to engage in a meaningful discussion before the unexpected discussion freeze. The primary concerns raised by the reviewers focus on limited novelty, reliance on perception module, and additional baselines. A key concern around novelty is that the method ‘largely recombines established ideas’ and fails to distinguish itself from existing HRL techniques. The authors attempt to address this via an extended discussion on HRL literature and an assertion that the integration of known techniques is the source of novelty. The reliance on perception is mitigated  with additional experiments that introduce noise in the perception module. Finally, experiments with some additional baselines are presented. These experiments do not provide evidence of improved performance from NEXUS compared to existing methods. Indeed, while the authors claim that NEXUS remains competitive in these comparisons, Table 3 seems to show that it falls short on multiple performance metrics compared to NUDGE and BlendRL. These results warrant a more detailed discussion of the performance-interpretability tradeoff of NEXUS, and additional experiments that would demonstrate the severity of this tradeoff. The authors also claim that ‘extending NEXUS to continuous spaces is theoretically straightforward’, but provide no justification for the lack of experiments on continuous control tasks. While they position this as future work, that framing would require some justification as to why it is not possible in the current paper, especially if such an extension is straightforward. Experiments on continuous control tasks become even more significant in light of the performance gap evidenced in Table 3.

**Reviewer Concerns:**

While concerns about novelty and reliance on perfect perception are largely addressed in the rebuttal, comparisons with additional baselines seems to show a significant performance gap between NEXUS and existing work. This performance gap is not discussed or contextualized in the revised manuscript.

**Reviewer Scores:**

None of the reviewers are likely to have changed their scores.

---

### Decision · Program_Chairs · 2026-01-26

Reject